# Involvement of circulating factors in the transmission of paternal experiences through the germline

Gretchen van Steenwyk[1,2,3,†] (ID), Katharina Gapp[2,3,4,5,6,7,†] (ID), Ali Jawaid[1,2,3,8], Pierre-Luc Germain[1,2,9] (ID), Francesca Manuella[1,2,3], Deepak K Tanwar[1,2,3,9] (ID), Nicola Zamboni[10], Niharika Gaur[1,2,3], Anastasiia Efimova[1,2,3], Kristina M Thumfart[1,2,3] (ID), Eric A Miska[5,6,7] (ID) & Isabelle M Mansuy[1,2,3,*] (ID)

## Abstract

Environmental factors can change phenotypes in exposed individuals and offspring and involve the germline, likely via biological signals in the periphery that communicate with germ cells. Here, using a mouse model of paternal exposure to traumatic stress, we identify circulating factors involving peroxisome proliferator-activated receptor (PPAR) pathways in the effects of exposure to the germline. We show that exposure alters metabolic functions and pathways, particularly lipid-derived metabolites, in exposed fathers and their offspring. We collected data in a human cohort exposed to childhood trauma and observed similar metabolic alterations in circulation, suggesting conserved effects. Chronic injection of serum from trauma-exposed males into controls recapitulates metabolic phenotypes in the offspring. We identify lipid-activated nuclear receptors PPARs as potential mediators of the effects from father to offspring. Pharmacological PPAR activation *in vivo* reproduces metabolic dysfunctions in the offspring and grand-offspring of injected males and affects the sperm transcriptome in fathers and sons. In germ-like cells *in vitro*, both serum and PPAR agonist induce PPAR activation. Together, these results highlight the role of circulating factors as potential communication vectors between the periphery and the germline.

**Keywords** blood serum; paternal experiences; PPAR; sperm; transmission
**Subject Categories** Chromatin, Transcription & Genomics; Metabolism; Neuroscience
**The EMBO Journal (2020) 39: e104579**

See also: **AJ Hannan** (November 2020)

## Introduction

Environmental factors and life events can have long-lasting consequences for exposed individuals, and in some cases, they can also impact their offspring. Transmission of environmentally induced features and diseases has been overlooked for decades. But today, evidence that diet, traumatic experiences or endocrine disruptors have effects across generations has accumulated in humans and experimental animals (Bohacek & Mansuy, 2015; Nilsson *et al*, 2018; Panzeri & Pospisilik, 2018). These effects are known to depend on epigenetic factors and constitute an important aetiological component of many diseases. When transmitted from parent to progeny and not depending on maternal care or social factors, they are thought to involve the germline. They therefore represent a form of heredity. But how exposure can affect the germline and which signals induced by exposure in the body can reach germ cells is not known. These signals may vary depending on the type of exposure, its time window, chronicity, etc. They have in common the ability to reach germ cells. Circulating factors are important vectors of communication between tissues and cells across the body. We postulate that they can carry signals induced by exposure to germ cells and contribute to the transmission of the effects of exposure to the progeny. Blood metabolites in particular are strong candidates for being such carriers because many are potent signalling molecules, e.g. hormones, lipids, organic acids and antioxidants. Further, they are dynamically regulated by physiological states in mammals. Several metabolites have been previously implicated in the epigenetic regulation of the genome in different tissues (Donohoe & Bultman, 2012; Kaelin *et al*, 2013; Sharma & Rando, 2017).

1  Laboratory of Neuroepigenetics, Brain Research Institute, Medical Faculty of the University of Zurich, Zurich, Switzerland
2  Institute for Neuroscience, Department of Health Sciences and Technology, ETH Zurich, Zurich, Switzerland
3  Zurich Neuroscience Center, ETH Zurich and University of Zurich, Zurich, Switzerland
4  Laboratory of Molecular and Behavioral Neuroscience, ETH Zurich, Zurich, Switzerland
5  Gurdon Institute, University of Cambridge, Cambridge, UK
6  Wellcome Trust Sanger Institute, Hinxton, UK
7  Department of Genetics, University of Cambridge, Cambridge, UK
8  Laboratory of Translational Research in Neuropsychiatric Disorders, BRAINCITY Nencki-EMBL Center of Excellence for Neural Plasticity and Brain Disorders, Warsaw, Poland
9  Statistical Bioinformatics Group, Swiss Institute of Bioinformatics, Zürich, Switzerland
10 Institute of Molecular Systems Biology, ETH Zurich, Zurich, Switzerland
   *Corresponding author. Tel: +41 44635 3360; E-mail: mansuy@hifo.uzh.ch
   †These authors contributed equally to this work

# Results

## Blood metabolites are altered in mice exposed to early life trauma

We examined the contribution of circulating metabolites to the effects of exposure from exposed individuals to their offspring. We used an established mouse model of early postnatal trauma based on unpredictable maternal separation combined with unpredictable maternal stress (MSUS) (Fig 1A and B). Mice exposed to MSUS have metabolic dysfunctions and behavioural deficits that are transmitted to the offspring across several generations (Franklin *et al*, 2010; Gapp *et al*, 2014b, 2016b; van Steenwyk *et al*, 2018). We conducted unbiased metabolomic analyses in exposed adult males and their offspring using time-of-flight mass spectrometry (TOF-MS). The analyses showed that polyunsaturated fatty acid (PUFA) metabolism, in particular, metabolites involved in α-linolenic/linoleic acid (ALA/LA), and arachidonic acid (AA) pathways are significantly upregulated by MSUS in plasma of adult males (Fig 1C). PUFAs, such as eicosapentaenoic acid (EPA) and dihomo-gamma-linoleic acid (DGLA), and arachidonic metabolites, such as the hydroxyeicosatetraenoic acids (HETEs), were the most significantly upregulated within the enrichments (Fig 1D). In addition, bile acid biosynthesis as well as steroidogenesis and the steroidogenic ligand aldosterone were downregulated (Fig 1C, full table in Appendix Figs S1 and S2). Altered steroidogenesis is consistent with previous observation in the MSUS model that the steroid mineralocorticoid receptor (MR) is downregulated. Its pharmacological blockade mimics some MSUS effects (Gapp *et al*, 2014b). Remarkably, except for AA metabolism, these pathways were also altered in the offspring of MSUS males when adult (Fig 1C, Appendix Fig S1). We focused the analyses on male offspring since bodyweight is significantly affected in male but not female offspring (Appendix Fig S3).

## Blood metabolites are altered in children exposed to early life trauma

We assessed the relevance of these results in humans by conducting similar analyses in a cohort of children (6- to 12-year-old girls and boys) from an SOS Children's Village in Lahore, Pakistan. The children have lost their father and were separated from their mother (paternal loss and maternal separation, PLMS) during the preceding year (Fig 1A). These conditions closely resemble the MSUS model. This human cohort is highly relevant for our study because SOS children have been exposed to a comparable trauma at a comparable age, which is key for correlative analyses with our mouse data. All SOS children live in the same orphanage, so differences in lifestyle factors are minimal. Control children were schoolmates living with both parents and not exposed to any trauma. PLMS and control groups were matched for age, body mass index and gender (Appendix Fig S4A–C). Control and PLMS groups attended the same school, with equal access to playground facilities and physical exercise. A Pakistani population was advantageous for this study because consanguinity is high in Pakistan (Bittles *et al*, 1991) (Appendix Fig S4D), making the group genetically more homogeneous than in other populations. Body mass index, diet and ethnicity account for < 5% of the variance between serum metabolites in healthy children from 6 different European populations (Lau *et al*,

2018), suggesting that our control samples are indeed comparable to other populations. Blood and saliva were collected from PLMS and control children. For these analyses and the following ones in humans, we used serum over plasma to avoid interference with clotting factors. Serum metabolites had significantly positive enrichment for AA metabolism and modest negative enrichment for bile acid biosynthesis compared to controls, with EPA, DGLA and HETEs being strongly affected (Fig 1C and D) similarly to MSUS. In saliva, both ALA/LA and AA metabolism, and steroidogenesis were also altered (Fig 1C, Appendix Fig S5), indicating alterations in different body fluids. Among the enrichments, individual metabolites in ALA/LA and AA pathways were comparably affected in MSUS and PLMS serum (Appendix Fig S6).

## PPAR is activated by MSUS blood metabolites

While we consider all circulating factors altered by MSUS to be potentially involved in germline transmission, we focused on the changes in fatty acids, especially PUFA, and their metabolites. PUFAs are known to modulate metabolism, inflammation and cognitive functions. Fatty acids and their metabolites can bind to various receptors, but are particularly potent ligands for peroxisome proliferator-activated receptors (PPARs). PPARs are widely expressed nuclear receptors that regulate gene expression and chromatin structure, and act by forming transcription factor complexes with retinoid X receptor (RXR). They can also interact with epigenetic modifying enzymes (Yu & Reddy, 2007; Romagnolo *et al*, 2014). Further to fatty acids, bile acids and steroid metabolites, also altered by MSUS, are ligands for nuclear receptors, in particular, farsenoid X receptor (FXR) and liver X receptor (LXR). FXR and LXR belong to the same family of nuclear receptors as PPAR and RXR, and can interact with them (Chawla *et al*, 2001).

Since PUFAs are PPAR ligands, we next examined if changes in circulating factors in MSUS males are linked to PPAR activity. We examined the expression of PPAR and some of their target genes in different tissues. In white adipose tissue, PPARγ is abundant and regulates adipocyte differentiation (Lee & Ge, 2014). PPARγ activation measured by transcription factor binding assay was increased in adult MSUS adipose tissue (Appendix Fig S7A). In liver, a tissue with high PPARα activity (Rakhshandehroo *et al*, 2010), several PPAR targets were differentially expressed, suggesting PPARα activation (Appendix Fig S7B). Further, because metabolic symptoms induced by MSUS are passed to the offspring, we also examined if germ cells have altered PPAR. PPARγ, the most abundant PPAR isotype in gametes (Aquila *et al*, 2006), was upregulated in MSUS sperm (Fig 2A). We then asked if metabolomic changes induced by MSUS can influence PPAR activity in germ cells. We used spermatogonial stem cell-like cells (GC-1 spg), diploid cells that resemble early-stage spermatogonial cells, to assess PPAR activity *in vitro*. Spermatogonial cells were chosen because they are the primary germ cells present in the developing testes at the time of MSUS (first 2 weeks after birth). GC-1 spg cells were exposed to culture medium enriched with 10% serum from control or MSUS adult males (Fig 2B). Prior to exposure, GC-1 spg cells were transfected with a plasmid expressing luciferase under the control of a PPAR response element (PPRE). Luciferase luminescence was significantly higher in cells exposed to MSUS serum compared to control serum (Fig 2C), indicating increased PPAR activation by MSUS serum.

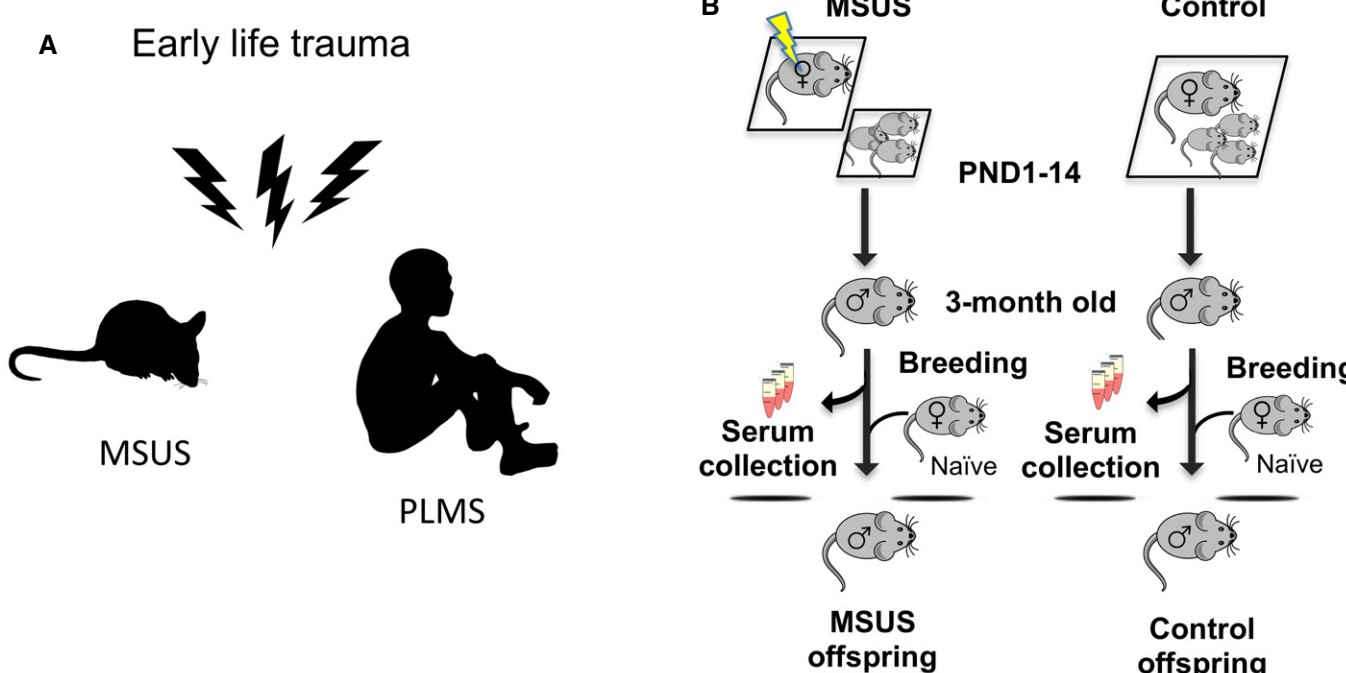

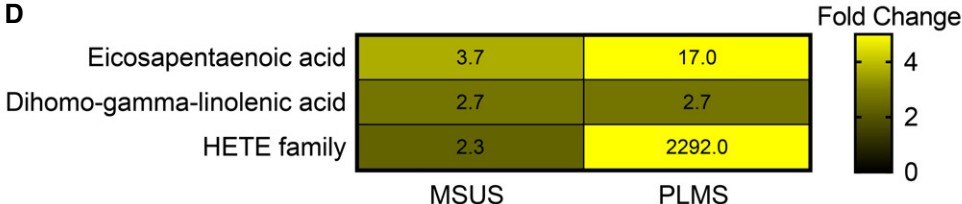

Upregulated metabolites (FDR < 0.05)

**C** Differential pathway enrichment table

| Enrichment | MSUS adults | | MSUS offspring | | PLMS | | PLMS (saliva) | |
|---|---|---|---|---|---|---|---|---|
| | - | + | - | + | - | + | - | + |
| Alpha linolenic and linoleic acid metabolism | / | **** | ** | * | / | / | / | *** |
| Arachidonic acid metabolism | / | **** | / | / | / | **** | / | **** |
| Bile acid biosynthesis | **** | / | **** | / | # | / | / | / |
| Steroidogenesis | # | / | * | / | / | / | / | ** |

**D**

| | MSUS | PLMS |
|---|---|---|
| Eicosapentaenoic acid | 3.7 | 17.0 |
| Dihomo-gamma-linolenic acid | 2.7 | 2.7 |
| HETE family | 2.3 | 2292.0 |

Fold Change scale: 0 – 2 – 4

**Figure 1. Effects of early life trauma on circulating metabolites in mice and humans.**

A  Paradigms of early life trauma in mice and humans. In mice, early life trauma consists of an exposure to unpredictable maternal separation combined with unpredictable maternal stress (MSUS). In humans, exposure involves paternal loss and maternal separation (PLMS) in early childhood (age 6–12).

B  Scheme illustrating the MSUS model and control mice showing blood collection and breeding to generate an offspring. For MSUS (symbolized by yellow blitz), newborn pups are separated from their mother unpredictably 3 h/day from postnatal day (PND) 1–14. During separation, the dam is exposed to different stressors unpredictably[9]. Serum was prepared from blood collected from 3-month-old MSUS and control males.

C  Differential pathway enrichment of metabolites in MSUS plasma from adult males and their offspring compared to controls (each group $n = 5$), and serum (PLMS, $n = 20$; control, $n = 14$) and saliva (PLMS, $n = 25$; control, $n = 14$) from PLMS and control children. Asterisk and hashtag represent FDR after multiple testing corrections using Benjamini–Hochberg (BH) test. Columns indicate significance for positive (+) and negative (−) enrichment.

D  Individual metabolites in ALA/LA and AA pathways significantly altered in both MSUS and PLMS. Numbers represent fold change according to a heat scale (right).

Data information: #FDR < 0.1, *FDR < 0.05, **FDR < 0.01, ***FDR < 0.001, ****FDR < 0.0001. (/) symbolizes non-significance. FDR, false discovery rate. ALA/LA, alpha-linolenic acid/linoleic acid. AA, arachidonic acid. HETE, hydroxyeicosatetraenoic acid.

## Paternal PPAR activation induces phenotypes into offspring and grand-offspring

Although previous studies have implicated PPAR and other nuclear receptors in the effects of environmental exposure and phenotype transmission (Lillycrop *et al*, 2008; Carone *et al*, 2010; Zeybel *et al*, 2012; Martínez *et al*, 2014; Baptissart *et al*, 2018), none have tested their causal involvement. We examined if PPAR can induce phenotype transmission by mimicking its activation in adult control males via chronic intraperitoneal (i.p.) injection of the dual PPARα/γ agonist tesaglitazar (10 μg/kg) (Fig 3A). We chose a dual agonist to activate multiple PPAR and better mimic the effects of MSUS. Following a 46-day delay after the last

injection to allow a full spermatogenesis cycle and eliminate transient effects of the drug, males were bred with control females to generate offspring. When adult, the offspring were bred with naïve females to produce grand-offspring. Both offspring and grand-offspring of tesaglitazar-injected males had significantly reduced body weight compared to the offspring and grand-offspring of vehicle-injected controls (Fig 3B; Appendix Fig S8A), despite an initial increase at PND8 (Appendix Fig S9). This effect was not due to a difference in fathers' weight after the injections or at the time of breeding (Appendix Fig S10). Further, blood glucose during a glucose tolerance test (GTT) was reduced in offspring and grand-offspring compared to controls (Fig 3C; Appendix Fig S8B), suggesting increased insulin sensitivity as

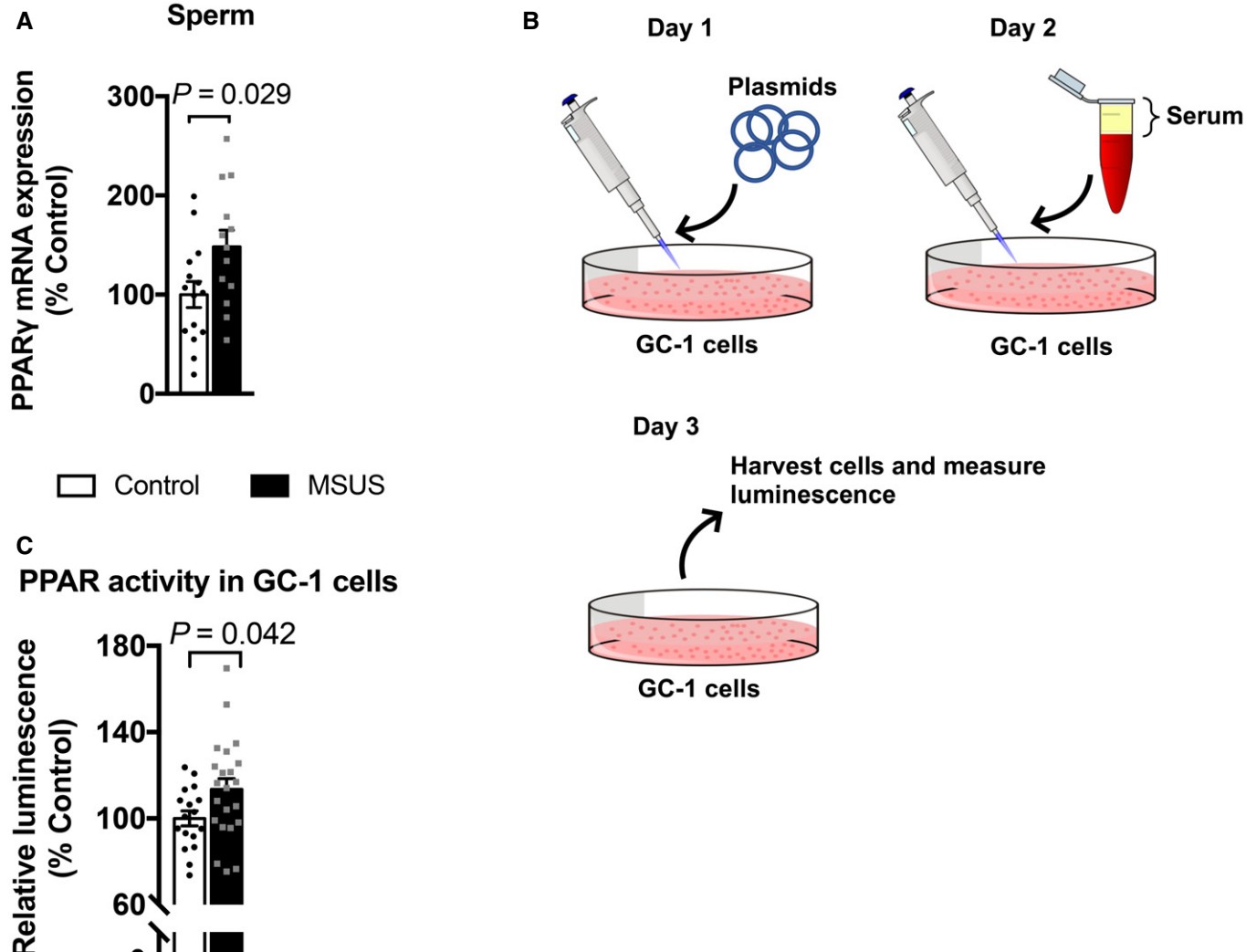

**Figure 2. Analyses of PPAR expression and PPAR activity in sperm and germ-like GC-1 cells.**

A  PPARγ mRNA expression in sperm from control and MSUS mice. Control, *n* = 15; MSUS, *n* = 13, two-tailed Student's *t*-test, *P* = 0.029, *t* = 2.30, df = 26.

B  Schematic of serum treatment of GC-1 cells. On day 1, cells are transfected with PPRE and luciferase normalization plasmids. Serum is mixed in culture medium at 10% concentration and exposed to cells on day 2. Cells are harvested 24 h later (Day 3), and luminescence is measured.

C  Relative luciferase luminescence in transfected GC-1 cells exposed to serum from control or MSUS mice. Numbers correspond to serum from individual animals applied to different cell culture wells. Control, *n* = 17; MSUS, *n* = 22, two-tailed Student's *t*-test, *P* = 0.042, *t* = 2.1, df = 37.

Data information: Data are reported as mean ± SEM.

previously observed in MSUS offspring (Gapp *et al*, 2014a). However, glucose response during a restraint stress was not altered, unlike in MSUS animals (Gapp *et al*, 2014a; Appendix Fig S11A). This may be because tesaglitazar does not involve the stress response unlike MSUS. These results indicate that tesaglitazar can mimic some of the metabolic effects of MSUS across generations.

We then asked if blockade of PPAR functions can have opposite effects in the offspring. Because PPARγ knockout models cannot produce viable offspring (Barak *et al*, 1999; Kubota *et al*, 1999) and lifelong alteration of PPAR activity strongly alters physiology (Bensinger & Tontonoz, 2008), we used a PPARγ antagonist (T0070907), the most suitable existing drug we could identify.

T0070907 or saline was injected in adult control males, and the injected males were bred to control females to obtain a progeny. PPARγ antagonist did not produce any effect on adult weight or blood glucose on GTT in the offspring (Appendix Fig S12A and B), suggesting that PPARγ inhibition in adulthood is not sufficient to affect metabolism or is effectively compensated for.

### MSUS-induced shift in sperm payload is reflected in sperm of tesaglitazar-injected fathers

Since sperm RNA has been causally involved in the transmission of the effects of MSUS to the offspring (Gapp *et al*, 2014a, 2020), we examined if RNA is altered in sperm of tesaglitazar-injected

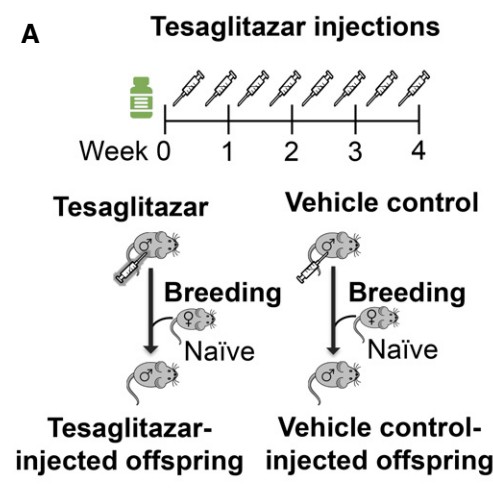

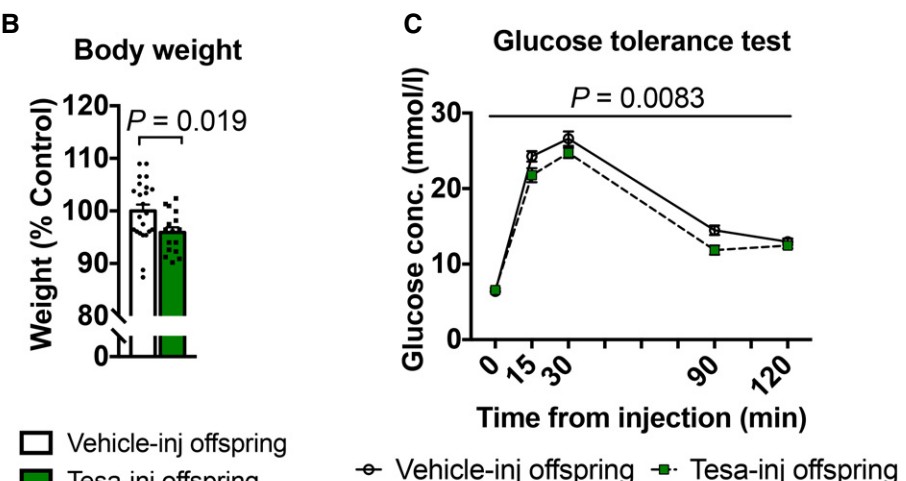

**Figure 3. Tesaglitazar injection reproduces MSUS phenotypes in the offspring.**

A   Control males were injected with either tesaglitazar (grey syringe) or vehicle twice per week for 4 weeks. Males were paired with control females to generate offspring.

B   Adult weight in the offspring of tesaglitazar-injected males (Tesa-inj, *n* = 16) compared to the offspring of vehicle-injected males (Vehicle-inj, *n* = 23). Two-tailed Student's *t*-test, *P* = 0.019, *t* = 2.44, df = 37.

C   Glucose level in the offspring of Tesa-inj (*n* = 14) and Vehicle-inj (*n* = 21) males during a glucose tolerance test. Repeated-measures ANOVA, treatment effect *P* = 0.0083, $F_{(1, 33)}$ = 7.877, time effect *P* < 0.0001, $F_{(4, 132)}$ = 347.7, interaction *P* = 0.0776, $F_{(4, 132)}$ = 2.155 Conc.; concentration.

Data information: Data are reported as mean ± SEM.

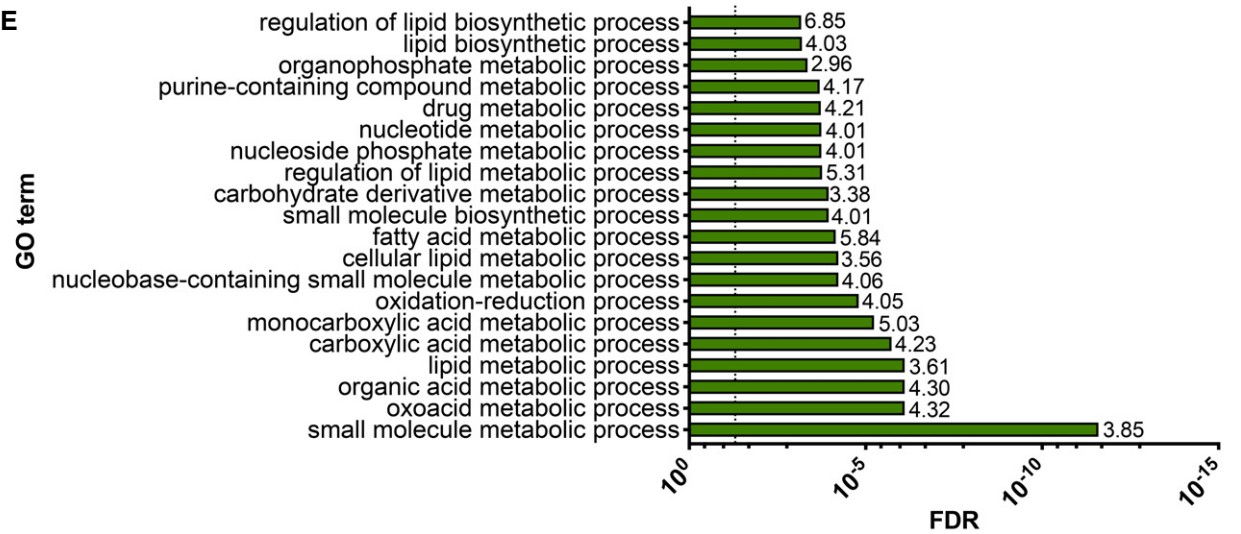

**Figure 4.**

**Figure 4. Differential RNA payloads in sperm from tesaglitazar-injected males and overlap with sperm from MSUS males.**

A, B   Differentially expressed (A) transposable elements and (B) mRNAs/lincRNAs in sperm from Tesa-inj and MSUS males. Data represent genes with $P < 0.05$ and similar fold change. Fold change in heat map represents log2(fold change) respective to the corresponding control group. Total overlap is presented in Appendix Fig S13.

C, D   (C) Volcano plot and (D) heat map of differentially expressed (FDR < 0.05) mRNA/lincRNA in sperm from Tesa-inj males. Dashed black lines in (C) represent $y = 1.3$, equivalent to FDR = 0.05, and $x = \pm 0.03$, equivalent to FC = 1.23 and 0.81. Black dots represent top candidates, which all have FC > 2 or FC < 0.5, and FDR < 0.05. Teal dots represent non-significant genes.

E   GO enrichment scores for differential RNA in Tesa-inj male sperm. Enrichments calculated with $P < 0.05$. For Tesa-inj, $n = 6$; Vehicle-inj, $n = 7$; MSUS, $n = 4$; and control, $n = 3$. Dashed black line represents FDR = 0.05. FDR, false discovery rate; FC, fold change.

males. Deep sequencing revealed differential RNA expression in sperm of tesaglitazar-injected males compared to vehicle-injected controls, in particular dysregulation of transposable elements (TEs) (Appendix Fig S13A). These results are consistent with previous observations in liver of tesaglitazar-treated mice (Ferguson *et al*, 2018) and in sperm of males exposed to MSUS (Gapp *et al*, 2020). In a targeted analysis, 630 TE and > 4,000 mRNA/lincRNA transcripts were annotated and there was a significant correlation in fold change of differentially expressed TEs (MSUS versus controls and tesaglitazar versus controls), including several long terminal repeat elements (LTRs) between tesaglitazar-injected and MSUS sperm (Fig 4A; Appendix Fig S13B). A modest fold change correlation of mRNAs/lincRNAs was also noted, in particular with lincRNAs that are the most enriched across datasets (Fig 4B; Appendix Fig S13C). Several mRNAs were also altered in sperm of tesaglitazar-injected males (FDR < 0.05), for instance, genes involving the mitochondrial respiratory chain complex (Fig 4C and D; Appendix Fig S14C), consistent with a role for PPAR in mitochondrial metabolism (Zhang *et al*, 2015). Further analysis of RNA in tesaglitazar-injected sperm confirmed significant GO term enrichments for fatty acid and lipid biosynthetic and metabolic processes (Fig 4E), pathways relevant for PPAR activity. Together, these data suggest a link between PPAR pathways in the periphery and long-term effects on sperm RNA.

### Serum upregulates PPAR activity in spermatogonial stem cell-like cells

RNA in sperm is thought to originate in part from earlier stages of spermatogenesis (Gapp *et al*, 2020), since mature sperm cells themselves are transcriptionally silent. Interestingly, sperm long RNA was not altered by tesaglitazar 1 day after the final injection unlike 46 days following the last injection (Appendix Fig S15A–D). This suggests that sperm cells are not directly affected (at least not at the level of RNA) and thus that the drug may induce changes at earlier spermatogenic stages that only appear later in mature sperm. To confirm this hypothesis, we assayed PPAR activity in GC-1 spg cells carrying a PPRE reporter exposed to serum collected 1 day after tesaglitazar injection and observed increased luminescence compared to cells exposed to serum from vehicle-injected controls (Appendix Fig S16). These results indicate that serum from both MSUS and tesaglitazar-injected males can upregulate PPAR activity directly in early-stage spermatogenic cells. Indirect effects of PPAR activation mediated by secondary factors present in circulation may, however, also occur *in vivo*. But this is unlikely the case with tesaglitazar since circulating metabolites are not affected by drug treatment, suggesting that there are no secondary factors at the time of breeding (Appendix Fig S17).

### *In vivo* serum transfer induces phenotype transmission to offspring

We next tested if serum can also have effects *in vivo*. Blood was collected from 4-month-old MSUS and control males, and serum was prepared and chronically injected intravenously (i.v.) in control adult males (Fig 5A). We chose serum over plasma for *in vivo* injections to avoid potential interference of clotting factors. Following 4-week treatment, males were bred with control females to generate offspring that were phenotyped when adult. The offspring of males injected with MSUS serum trended towards reduced weight (Fig 5B) and had significantly decreased blood glucose upon acute stress (Fig 5D), similar to that observed in MSUS offspring (Fig 5B and C, Gapp *et al*, 2014a). There was no difference in blood glucose on GTT in these offspring (Appendix Fig S11B), like in MSUS mice (Gapp *et al*, 2014a). The results indicate that the metabolic effects of MSUS serum are different from those of tesaglitazar treatment, which is expected since MSUS is a complex paradigm that activates components of the stress response, such as hormones, non-lipid metabolites and other circulating and cellular constituents. To assess whether other factors than metabolites are involved, we examined proteins in plasma using unbiased mass spectrometry. No significant difference passing multiple testing corrections could be detected (Appendix Fig S18A). One interesting candidate was C-reactive protein (CRP), which was downregulated in the proteomic datasets and could be confirmed by ELISA in a separate batch of MSUS samples (Appendix Fig S18B). CRP is a marker of inflammation linked to PPAR signalling and can be negatively regulated by PPAR activity (Zambon *et al*, 2006). We also examined RNA in serum since circulating miRNAs are known to communicate with tissues outside their site of origin (Thomou *et al*, 2017), and because RNA itself has been implicated in epigenetic inheritance (Rassoulzadegan *et al*, 2006; Gapp *et al*, 2014a; Grandjean *et al*, 2016). No significant difference in small RNAs could be detected in MSUS serum after multiple testing correction (Appendix Fig S19). These results suggest that circulating miRNAs probably do not play a major role, although exosomal or HDL-associated RNA uptake (Cossetti *et al*, 2014) cannot be excluded. It should be noted that individual miRNAs were previously found by qPCR to be altered in MSUS serum (Gapp *et al*, 2014a) but this discrepancy may be due to technical differences in serum preparation or in RNA detection between RNA sequencing and qPCR.

## Discussion

Germ cells are the carrier of biological heredity that passes information from parent to progeny, information that is now recognized to

involve both the genome and the epigenome (Chen *et al*, 2016). Because germ cells are sensitive to environmental factors, especially in early life (Day *et al*, 2016), they are subjected to alterations by exposure. If these alterations persist and are present at the time of conception, they may be transferred to the offspring. Our results (Appendix Fig S20) newly identify circulating factors as causal

mediators of metabolic effects of postnatal trauma from exposed father to the offspring and highlight PPAR as one of the molecular contributors. While stress activates the sympathetic nervous system and the hypothalamic–pituitary–adrenal (HPA) axis, our results show that lipid metabolism also plays a role in the response of the body to stress exposure, and implicate PPAR beyond dietary insults

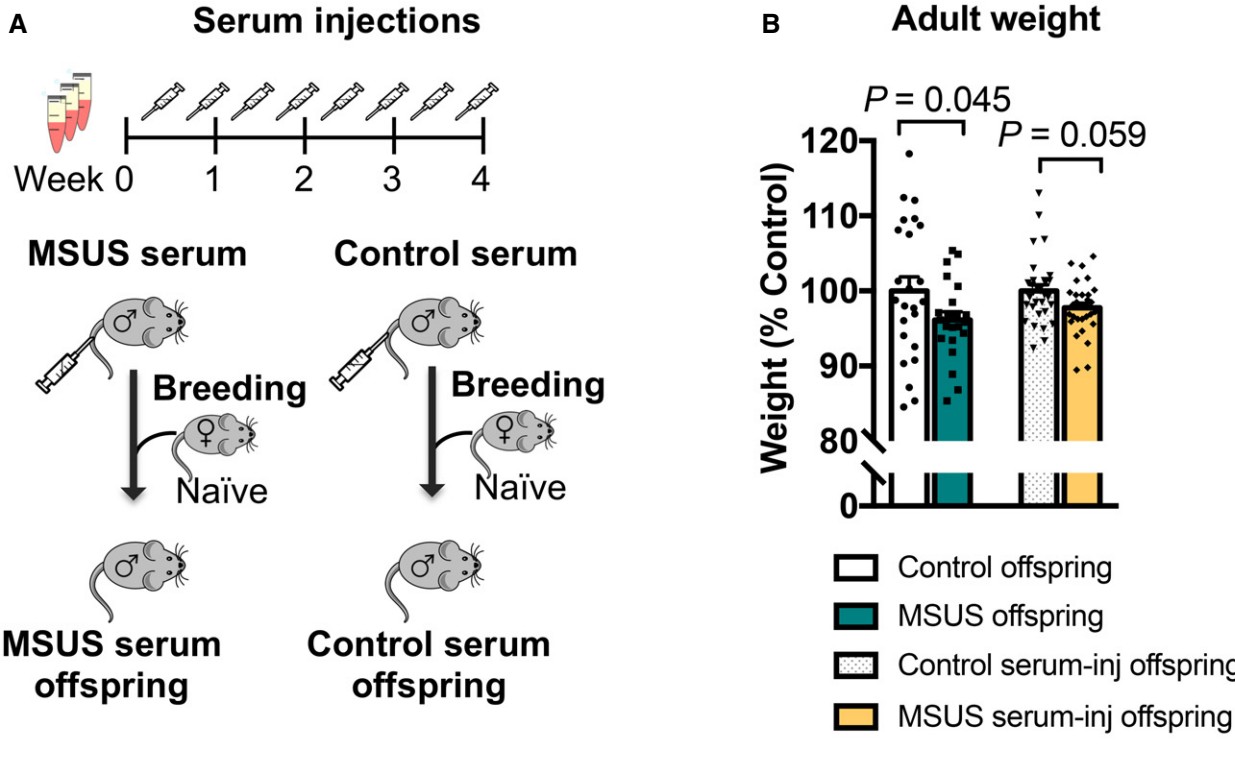

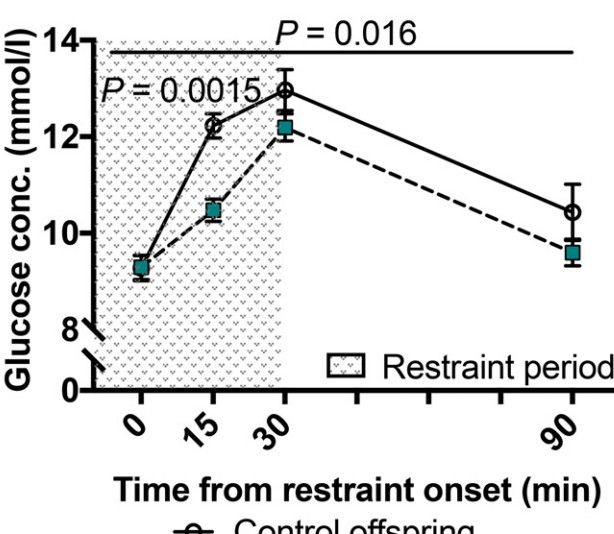

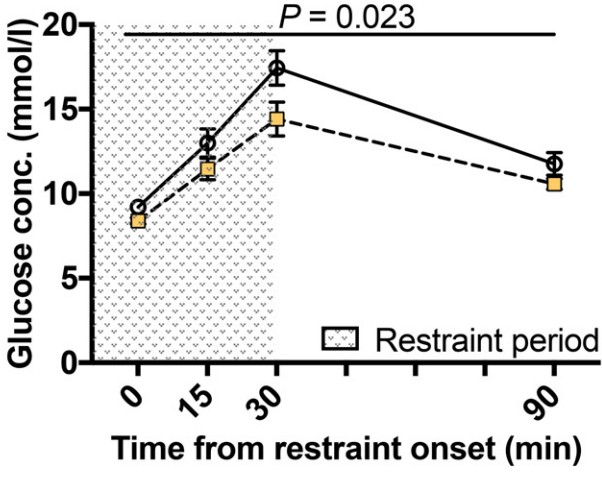

**Figure 5.**

**Figure 5. Injection of serum from MSUS mice recapitulates some MSUS metabolic phenotypes in the offspring.**

A    Serum was injected (90 µl) twice per week for 4 weeks into age-matched control males. After injection, the males were paired with control females and their offspring were phenotyped when 3 months old and compared to the offspring of MSUS males.

B    Weight in adult male MSUS offspring and the offspring of males injected with MSUS serum compared to respective control groups. MSUS offspring, $n = 22$; control offspring, $n = 24$, one-tailed Student's $t$-test (data reproduced) $P = 0.045$, $t = 1.734$, df = 44. MSUS serum-injected offspring, $n = 30$; control serum-injected offspring, $n = 31$, two-tailed Mann–Whitney $U = 334.5$, $P = 0.059$.

C, D  Blood glucose levels in MSUS offspring and the offspring of MSUS serum-injected males following a 30-min restraint challenge. MSUS offspring, $n = 13$; control offspring, $n = 12$, repeat-measures ANOVA, treatment effect $P = 0.016$, $F_{(1, 23)} = 6.704$, time effect $P < 0.0001$, $F_{(3, 69)} = 53.13$, interaction $P = 0.027$, $F_{(3, 69)} = 3.25$, at 15 min adjusted $P = 0.0015$, $t = 3.693$, df = 92. MSUS serum-injected offspring, $n = 17$; and control serum-injected offspring, $n = 14$, repeat-measures ANOVA, treatment effect $P = 0.023$, $F_{(1,14)} = 6.493$, time effect $P < 0.0001$, $F_{(3, 42)} = 48.4$, interaction $P = 0.29$, $F_{(3, 42)} = 1.29$. Conc., concentration.

Data information: Data are reported as mean $\pm$ SEM.

(Carone *et al*, 2010; Zeybel *et al*, 2012; Chamorro-Garcia *et al*, 2017). This may have important implications for the clinic since metabolic syndrome is a common co-morbidity in the sequela of childhood trauma (Suglia *et al*, 2018). The results also provide a novel link between PPAR and germ cells, and a possible biological role for the known property of transcription factors to poise transcriptional states in gametes and influence the developmental trajectory of zygotes (Jung *et al*, 2017). The additional correlation between PPAR activation and MSUS-induced TE dysregulation in sperm further points towards potential consequences on epigenetic control of TE-coregulated genes in the embryo (Sharma *et al*, 2016). PPAR pathways in germ cells may thus contribute to differential gene expression previously observed in MSUS offspring at zygotic stage (Gapp *et al*, 2020). Serum factors may also act in part via somatic cells in gonads such as Sertoli, Leydig or epididymal cells which have direct contact with germ cells (Sæther *et al*, 2007; Sharma *et al*, 2016). Although the contribution of such intermediate cellular signals between circulation and germ cells cannot be ruled out, our results indicate that factors in serum are sufficient to induce the transmission of phenotypes to the offspring. The contribution of extracellular vesicles to the distribution of altered metabolites or additional factors such as RNA relevant for transmission has not been assessed but could present another potential vector for information transfer. Other mechanisms involving epigenetic processes, already implicated in phenotype transmission in several models including MSUS (Franklin *et al*, 2010; Gapp *et al*, 2014b), may also operate. It would be interesting in the future to conduct metabolomic profiling in other inter- or transgenerational models and in other body fluids such as lymph or seminal fluid (Di Venere *et al*, 2018) to clarify the role of such pathways in transmission. Moderate alterations in single metabolites are unlikely to induce transmission of the observed phenotypes on their own, yet it might be interesting to evaluate whether a large excess of individual metabolites and activation of target receptors could induce a phenotype in the offspring. Notably, our results suggest beneficial effects of tesaglitazar on offspring metabolism as reported earlier (Wallenius *et al*, 2013), which is consistent with the known therapeutic benefit of PUFA supplementation (Huber *et al*, 2007; Braarud *et al*, 2018). The timing and duration of PPAR activation may also be further optimized to elicit differential effects.

Finally, these findings bring new fundamental knowledge about the influence of the environment on the germline by extending the notion of non-DNA based inheritance, known to involve epigenetic processes, to circulating factors, with important implications for heredity and evolution.

# Materials and Methods

## Mice

C57Bl/6J mice were kept under a 12-h reverse light/dark cycle in a temperature- and humidity-controlled facility. Animals had access to food and water *ad libitum*. Experimental procedures were performed during the animals' active cycle (reverse light cycle in the facility, light on at 9 am and off at 9 pm) in accordance with guidelines and regulations of the Cantonal Veterinary Office, Zürich, except for those involving serum-injected mice and their offspring that were performed during the animals' inactive cycle as approved by the Home Office, United Kingdom. Animal licences covering this work have the number: 57/15 and 83/18 cover the conducted experiments.

## Msus

To obtain MSUS mice, 3-month-old C57Bl/6J primiparous females were paired with age-matched control males for 1 week; 40 breeding pairs were used. After birth of pups, dams were randomly assigned to MSUS or control groups, in a way to balance litter size and number of animals across groups. Dams assigned to MSUS group were separated from their pups for 3 h per day unpredictably from postnatal day (PND) 1 to 14. Separation onset was at an unpredictable time within the 3 h, and during separation, each mother was randomly exposed to an acute swim in cold water (18°C for 5 min) or 20-min restraint in a tube. Control animals were undisturbed apart from cage changes once per week (like MSUS). At PND21, pups were weaned from their mother and assigned to cages in groups of 3–5 mice/cage housed by gender and treatment. Siblings were distributed in different cages and mixed with pups from different mothers to avoid litter effects. Total number of fathers and offspring for each experiment is provided in Appendix Fig S21.

## Metabolomic measurements

Metabolites were extracted from 10 µl plasma 3 times with 70% ethanol at a temperature > 70°C. Extracts were analysed using flow injection–time-of-flight mass spectrometry (Agilent 6550 QTOF) operated in negative mode, as described previously (Fuhrer *et al*, 2011). Distinct mass-to-charge (m/z) ratio could be identified in each batch of samples (typically with 5,000–12,000 ions). Ions were annotated by aligning their measured mass to compounds defined by the KEGG database, allowing a tolerance of 0.001 Da. Only

deprotonated ions (without adducts) were considered in the analysis. When multiple matches were identified, such as in the case of structural isomers, all candidates were retained. For enrichment analysis, metabolites with $P < 0.05$ and log2(fold change) $> 0.25$ or $< -0.25$ follow a previously described procedure (Subramanian *et al*, 2005). Enrichments were considered significant when FDR $< 0.05$ after multiple testing corrections using the Benjamini–Hochberg post hoc test.

## Blood and tissue collection

Mouse pup siblings at postnatal day 8 (after 7 days of MSUS treatment) were sacrificed from each cage after removing the mother, and all pups (up to 10) were sacrificed within 2 min. For blood and tissue collection in adults, males were singly housed overnight with food and water to avoid activating their stress response by the successive removal of littermates. Details of serum/plasma processing are described in different sections of the methods.

### Blood collection for metabolomic and proteomic analyses

Trunk blood was collected in EDTA-coated tubes (Microvette, Sarstedt) from PND8 pups (5 control and 5 MSUS, from different litters) and 4-month-old (for MSUS offspring, 5 control and 5 MSUS, from different litters) or from 6-month-old (for MSUS adults, 5 control and 5 MSUS, from different litters) mice after decapitation. Samples were stored at 4°C for 1–3 h and centrifuged at 2,000 × *g* for 10 min. Plasma was collected and stored at −80°C until processed for analysis. For analyses, samples from pups from different litters (and not from siblings) or from adult mice from different cages (and not cage mates) were chosen.

### Sperm collection

Sperm was collected using a swim-up method as described previously (Brykczynska *et al*, 2010). Briefly, cauda epididymis was dissected and perforated with dissection scissors and placed in M2 medium (Sigma-Aldrich) for 1 h to allow mature sperm to swim out of the tissue. Medium was collected and spun down at 2,000 × *g* for 6 min at 4°C. Sperm pellets were then resuspended in 15 ml somatic cell lysis buffer (contains 1% SDS (10%) and 0.5% Triton X-100 in Milli-Q water) and left to incubate on ice for 10 min. Sperm was re-pelleted by centrifugation at 2,000 × *g* for 6 min at 4°C and washed twice with ice-cold PBS (spun after each wash at 2,000 × *g* for 6 min at 4°C). After the final wash, an aliquot of sperm was used for counting (Appendix Fig S22) and then pellets were stored at −80°C until further use.

### Tissue collection

Following decapitation and blood collection, mice were pinned by their feet to a sterilized dissection board. A midline incision was made from the subcostal region to the pubic zone where two transverse incisions were made bilaterally to the femur, exposing internal organs. Biopsies from white adipose tissue were collected proximal to the epididymis on each side.

## Aldosterone enzyme-linked immunosorbent assay (ELISA)

Aldosterone concentration in serum was measured using a competitive ELISA (Abnova, KA1883) according to the manufacturer's protocol. 50 µl of standards, control and serum samples from adult MSUS and control males was added to wells containing aldosterone antibody, and an HRP-conjugated aldosterone antigen was added. After 1-h incubation at room temperature for competitive binding, cells were washed and tetramethylbenzidine (TMB) substrate was added for colour development, followed by addition of stop solution. Absorbance was measured at 450 nm with a NOVOstar Plate Reader (BMG Labtech), and sample absorbance (inversely related to concentration) was determined by plotting a standard curve with manufacturer-supplied standards and controls. Samples were measured in duplicate and averaged.

## Human cohort

To assemble a cohort of children exposed to trauma, we contacted the administration of SOS Children's Village in Lahore, Pakistan, and selected children using the following criteria at the time of assessment: (i) age between 6 and 12 years, (ii) paternal death, (iii) maternal separation in the form of adoption by the SOS village and (iv) entry of the child to the SOS village within 12 months preceding the assessment. Maternal separation was forced because mothers could no longer provide sufficient support to their child and had to transfer their care to the SOS village. They had no or minimal contact with their child at the time of assessment. Maternal suffering during childhood and early adolescence is known to affect mental health in adulthood (Abel *et al*, 2014). Paternal loss was used as an inclusion criteria, because spousal death is a critical life stressor in human (Prior *et al*, 2018), and serves to mimic unpredictable maternal stress of the MSUS model. Exclusion criteria included: (i) history of abuse and (ii) history of traumatic brain injury, intellectual disability or cerebral palsy. Based on these criteria, a total of 26 children with paternal loss and maternal separation (PLMS) were selected. A control group ($n = 16$) was recruited among schoolmates of PLMS children and comprised 6- to 12-year-old children living with both parents and having no history of trauma, traumatic brain injury, intellectual disability or cerebral palsy. Complete confidentiality of participants was maintained at all stages of data collection and analyses. The administration of the SOS village, Lahore, Pakistan, was informed and approved all study procedures.

### Demographics

Detailed demographic information for PLMS and control groups was provided by the administration of the SOS village for the PLMS children and from parents for control children. This included age, gender, parental consanguinity (defined by 1st or 2nd cousin parental union) and physical health records. Weight and height were measured in all children by two research interns blinded to the study design. Children were classified as underweight, healthy weight or overweight based on their correspondence to the "less than 5th percentile", "between 5th and 85th percentile" and "85th to 95th percentile" reference ranges defined for Pakistani children of the same age and sex (Mushtaq *et al*, 2012).

### Serum and saliva sample collection

Blood was collected by a trained phlebotomist blinded to the study design. Blood withdrawal took place during the morning school hours for both groups approximately 1 h after their last oral intake. All children received a brief explanation of the blood withdrawal procedure and were promised a gift basket for their cooperation.

                                                   

Children showing reluctance or despair were excluded (6 PLMS and 2 controls). After sterilization with a swab, axillary vein venepuncture was performed through a butterfly syringe and 6 ml blood was collected per child in serum separating tube (BD Vacutainer, Thermo Fisher scientific). After 1-h incubation at room temperature, the tubes were centrifuged at $1,300 \times g$ for 10 min at 4°C for serum separation. Extracted serum was aliquoted into 1.5-ml tubes and stored at −80°C. Saliva was collected by two research interns blinded to the study design. Children who had active upper respiratory tract infections (identified through the symptoms of fever, rhinorrhoea or cough) at the time of sample collection were excluded (1 PLMS and 2 controls). All children received a brief explanation of the saliva collection procedure and were promised a gift basket for their cooperation. After a 1-h period of no oral intake, the children were asked to rinse their mouth with clear water twice. Saliva was collected 5 min after rinsing through passive drooling in salivette tubes (Sarstedt) over a period of 5 min. Collected saliva was aliquoted into 1.5-ml tubes and stored at −80°C. Serum and saliva aliquots were shipped to Zürich in packages containing 20 kg of dry ice for analysis.

### Luciferase assay in GC-1 spg cells

GC-1 spg cells were obtained from ATCC (ATCC® CRL-2053™) and cultured at 37°C in Dulbecco's modified Eagle's medium (DMEM—high glucose, Sigma-Aldrich) supplemented with 10% (v/v) foetal bovine serum (FBS, HyClone) and 40 µg/ml gentamicin (Sigma-Aldrich). Cells were passaged 1:10 every 3–4 days for 3 passages before being transfected. Prior to transfection, 4,000 cells were plated per well in 48-well plates. Cells were co-transfected with PPRE X3-TK-luc plasmid, expressing firefly luciferase, and pRL-SV40P plasmid, expressing Renilla luciferase, using Lipofectamine 2000 Reagent (Thermo Fisher Scientific), according to the manufacturer's protocol. PPRE X3-TK-luc was a gift from Bruce Spiegelman (Kim *et al*, 1998) (Addgene plasmid #1015; http://n2t.net/addgene: 1015; RRID:Addgene_1015), and pRL-SV40P was a gift from Ron Prywes (Chen & Prywes, 1999) (Addgene plasmid #27163, RRID: Addgene_27163). Plasmid DNA and Lipofectamine reagent were mixed in the supplied OptiMEM and added to culture medium such that each well received 200 ng plasmid DNA and 0.5 µl Lipofectamine. Following 24 h, transfection medium was replaced with control or MSUS serum-enriched medium. Serum was added to culture medium at 10% (v/v) and then sterile-filtered using 0.22-µm PVDF filter units (Merck) before being dispensed into individual wells. Each cell sample was treated with serum collected from an individual adult male. Luciferase signal produced from firefly and Renilla reporter plasmids was measured using the Dual Reporter Luciferase Assay System (Promega) with a GloMAX Multi Detection System (Promega). For each sample, firefly luminescence was normalized to the stable Renilla luminescence signal coming from the same well. Firefly and Renilla luminescence signals were absent in non-transfected cells (Appendix Fig S23).

### RT–qPCR

For gene expression analyses in sperm and liver, RNA was extracted using a phenol/chloroform extraction method (TRIzol; Thermo Fisher Scientific). Reverse transcription was performed on purified RNA samples with miScript II RT reagents (Qiagen) using HiFlex buffer. RT–qPCR was performed with QuantiTect SYBR (Qiagen) on a Light Cycler II 480 (Roche). All samples were run in triplicate under the following cycling conditions: 95°C for 15 min, 45 cycles of 15 s at 94°C, 30 s at 55°C and 30 s at 70°C, followed by melt curve with gradual temperature increase until temperature reached 95°C. Melt curve analysis confirmed amplification of single products for each primer. The endogenous control TUBD1 for sperm, and RPLP0 for liver, was used for normalization. Primer sequences are proprietary (Qiagen). Expression levels were analysed with two-tailed Student's *t*-test.

### PPARγ transcription factor binding assay

PPARγ transcription factor binding activity was measured in nuclear extracts from epididymal white adipose tissue (eWAT). Nuclear proteins were collected from 10 mg eWAT using a Nuclear Extraction Kit (Abcam, ab113474). Protein concentration in nuclear extracts was measured using Qubit Protein Assay Kit (Thermo Fisher Scientific), and volumes were adjusted to have similar protein concentration across samples. PPARγ binding activity was measured using a PPARγ Transcription Factor Assay Kit (Abcam, ab133101) following the manufacturer's protocol. Briefly, 10 µl of nuclear extracts was added with binding buffer to individual wells of a plate conjugated with DNA sequences containing known PPARγ binding motifs and incubated at 4°C overnight. Wells were washed 5× with wash buffer, and primary antibody against PPARγ was added for 1 h at room temperature without agitation. Wells were washed again 5× with wash buffer, and an HRP-containing secondary antibody was added to wells and incubated at room temperature for 1 h. Following another 5× wash, a developing solution was added and incubated under gentle agitation for 45 min. The reaction was stopped by adding stop buffer, and absorbance was measured immediately with NOVOstar Plate Reader (BMG Labtech). Samples were measured in duplicate, alongside a positive control supplied by the manufacturer to confirm assay success. Data were analysed with two-tailed Student's *t*-test.

### Drug injections

#### *Tesaglitazar*
Three-month-old mice were injected i.p. with either tesaglitazar solution at 10 µg per kg body weight or vehicle control twice per week for 4 weeks. Tesaglitazar (Sigma-Aldrich) was solubilized in DMSO at 10 µg per µl and resuspended in sterile 0.9% saline. Vehicle consisted of equivalent DMSO in 0.9% sterile saline. Mice were weighed at baseline, before each injection and 6 weeks after the last injection. Males were paired with 3-month-old primiparous control females 46 days after the last injection. Pairing lasted 1 week; then, males were grouped back with their original littermates. After pairing, females were separated into individual cages until delivery. The offspring were reared in standard conditions, with one cage change per week, and were weaned at PND21 in social groups (3–4/cage) with pups from other litters to avoid litter effects. When 3 months old, offspring were subjected to phenotyping.

#### *T0070907*
3-month-old control and MSUS males were injected with either vehicle control or 1.5 mg per kg T0070907 once per week for 2 weeks.

T0070907 was solubilized in DMSO at 10 µg per µl and resuspended in 0.9% saline; vehicle control consisted of 0.9% saline with equivalent DMSO concentration. Mice were paired with females 1 day after the final injection. Pairing lasted 1 week; then, males were placed back in cages with original littermates. After pairing, females were separated into individual cages until delivery. The offspring were reared in standard conditions, with one cage change per week, and were weaned at PND21 in social groups (3–4/cage) with pups from other litters to avoid litter effects. When 3 months old, offspring were subjected to phenotyping.

## Metabolic testing

Before testing, cages were labelled such that the experimenter was blind to treatment group. When more than one experimenter was required to conduct a given test, each experimenter tested a similar number of control and MSUS animals to exclude experimenter-specific effects. Control and MSUS treatment groups were tested alternately or side by side to avoid circadian effects (in a blinded manner for the experimenter). All glucose measurements were performed using fresh blood droplets with an Accu-Chek Aviva glucometer (Roche).

### Glucose in response to restraint
Mice were single-housed for minimum 4 h but no more than 18 h prior to testing with access to food and water. For physical restraint, each mouse was confined individually in a cylindrical plastic tube (3.18 cm diameter with sliding nose restraint, Midsci) for 30 min. Blood was drawn at 0, 15, 30 and 90 min after initiation of restraint using a 28-G needle to prick the tail from within 1 cm of the tip. After blood was collected at the 30-min time point, each mouse was released and placed in an individual temporary cage. At 90 min, the mouse was briefly (10 s) placed under an inverted 1-litre glass beaker (dimensions: 14.5 cm high and 12 cm diameter) with its tail positioned to protrude from the beaker spout for easy access by the experimenter. The mouse was then placed back into its temporary cage for 1 h and then returned to its original group cage. Data were analysed with repeated-measures ANOVA and corrected for multiple comparisons using Šidák post hoc test.

### Glucose tolerance test
Mice were singly housed without food starting between 5 and 6 pm, and testing began at 9am the next morning. Glucose was measured in blood samples at 0, 15, 30, 90 and 120 min following intraperitoneal (i.p.) injection of sterile glucose solution containing 2 mg per g of body weight in 0.45% (wt/vol) saline. Each mouse was kept under an inverted 1-litre beaker with its tail in the spout as described above. After taking the 30-min measurement, each mouse was placed in an individual cage and then taken again out briefly (10 s) for measurements at 90- and 120-min time point. The mouse was placed back into its temporary cage for 1 h before returning to its original group cage, to reduce fighting due to experimental stress. Data were analysed with repeat-measures ANOVA and corrected for multiple comparisons using Šidák post hoc test.

### Food intake and weight measurement
All animals were weighed using the same scale at the same time of day. Following weight measurements, food intake was also measured. Food pellets were weighed at the beginning and end of three consecutive days and replaced every 24 h to limit crumb spillage. Food consumption was calculated per cage (maximum 5 animals) and averaged per animal. No difference in food intake was observed. Data were analysed with two-tailed Student's *t*-test, except for MSUS offspring, which was analysed with one-tailed Student's *t*-test to confirm previous reports (Gapp *et al*, 2014a).

## RNA sequencing

RNA was extracted from sperm using the TRIzol/chloroform method and analysed using Bioanalyzer (Agilent 2100). Sequencing was performed using Illumina Genome Analyzer. RNA libraries were prepared with the TruSeq Small RNA and TruSeq Stranded Total RNA kits according to the manufacturer's instructions with the following modifications: 1:3 for sperm long RNA. Serum small RNA was amplified for a total of 16 cycles, and sperm long RNA was amplified for a total of 15 cycles. 220 ng total sperm RNA was depleted of rRNAs using Ribo-Zero Gold and further processed in RNA libraries. High-throughput sequencing was performed on a Genome Analyzer HiSeq 2500 (Sanger Institute, Cambridge) for 36 and 51 cycles for 50 bp and 100 bp runs, respectively, plus 7 cycles to read the indexes. Serum small RNA libraries were run twice, and data were merged. For library preparation of sperm RNA collected from males 24 h after tesaglitazar injection, a slightly different protocol was used. The quality of the isolated RNA was determined with a Qubit® (1.0) Fluorometer (Life Technologies, California, USA) and a Fragment Analyzer (Agilent, Santa Clara, California, USA). TruSeq Stranded mRNA kit (Illumina, Inc, California, USA) was used to prepare libraries. Briefly, total RNA (300 ng per sample) was polyA-enriched and then reverse-transcribed into double-stranded cDNA. The cDNA was fragmented, end-repaired and adenylated before ligation with TruSeq adapters containing unique dual indices (UDI) for multiplexing. Fragments containing TruSeq adapters on both ends were selectively enriched by PCR. The quality and quantity of enriched libraries were validated using Qubit® (1.0) Fluorometer and Fragment Analyzer (Agilent, Santa Clara, California, USA). The product is a smear with an average fragment size of approximately 260 bp. The libraries were normalized to 10 nM in Tris–Cl 10 mM, pH 8.5, with 0.1% Tween-20; then, NovaSeq 6000 (Illumina, Inc, California, USA) was used for cluster generation and sequencing according to a standard protocol. Sequencing was paired end at 2 × 150 bp or single end at 100 bp.

### Long RNA-sequencing analysis
Single-end reads were assessed for quality using FastQC (version 0.11.5) (Andrews, 2010). Quality control was performed with Trim Galore (version 1.16) (Krueger, 2012), and bases with quality score < 30 (-q 30) and reads shorter than 30 bp were also removed (–length 30), and adapters were removed. Filtered reads were pseudo-aligned with Salmon (version 0.11.2) (Patro *et al*, 2017) with library-type parameter (-l SR), on a transcriptome index prepared with (i) the GENCODE (Frankish *et al*, 2019) annotation (version M18), (ii) piRNA precursors (Li *et al*, 2013) and (iii) transposable elements (TEs) from repeat masker (concatenated by family). For differential expression analysis, normalization factors were calculated using the TMM method (Robinson & Oshlack, 2010) on all quantified RNAs aggregated at the gene (or repeat element family)

level, and repeat elements or mRNA and lincRNAs were selected for testing. Only features with more than 20 reads in at least a number of samples equivalent to 80% of the size of the smallest experimental group were considered. In the case of MSUS sperm, where two libraries per sample were available, the voom/dupCor method of the limma R package v.3.34.9 (Ritchie *et al*, 2015) was used as previously (Gapp *et al*, 2020) to account for non-independence of the samples. In all other cases, edgeR v.3.24.0 was used with the exact test. Concordance between datasets was estimated through a Pearson correlation of fold change of genes with $P < 0.05$ in both datasets. Gene Ontology (GO) enrichment analysis was performed using the goseq R package (Young *et al*, 2010) (Fisher's exact test). This accounts for length bias in RNA-seq experiments using GO terms with 10–1,000 annotated genes. Only genes with existing GO annotations were used. Only differentially expressed genes subjected to count filtering were used as background for enrichment analysis. Since duplication can be attributed to biological variations, duplicates were not removed from the analyses.

### Short RNA-sequencing analysis

RNA-seq reads were trimmed of adapter sequences using cutadapt v1.14 with a 5% error rate and collapsed to unique sequences before being aligned on a custom genome containing, in addition to normal contigs, DNA sequences of spliced post-transcriptionally modified RNAs (such as tRNAs) obtained from GtRNAdb 2.0 database. Reads were first aligned using bowtie1 end-to-end alignment without mismatch (and accepting -m 1000), and remaining reads were aligned allowing soft clipping and a mismatch. Reads overlapping with known features (including miRBase miRNAs, GENCODE and RepeatMasker features, and piRNA precursors) were counted, resolving multiple alignments through a hierarchy of features (for instance, reads mapping to multiple tRNA genes of the same family were assigned to the family). Normalization factors were calculated over ll identified features using the TMM method, and differential expression was performed at all levels of the feature hierarchy.

### Serum sampling and intravenous injection

Four-month-old MSUS and control males were sacrificed by decapitation, and trunk blood was collected in non-coated Eppendorf tubes and clotted overnight at 4°C. After centrifugation for 10 min at $2,000 \times g$ at 4°C, serum was collected and stored at $-80$°C. When 2 months old, control mice received 8 tail vein injections of 90 μl of serum from adult MSUS or control mice over the course of 4 weeks (twice/week). For each group, serum from 43 mice was pooled before injections to obtain sufficient volume for all injections. For injections, mice were placed in a tube in a heating chamber at 38°C. Injections were alternated between opposing lateral veins of the tail. Proper insertion of the injection needle was successful on the first attempt in most cases and never took more than 3 attempts. Each male was paired with a primiparous adult control female 12 days after the last injection for 1 week.

### Proteomic measurements

Plasma samples were enriched for small regulatory proteins using protein depletion columns (Seppro Mouse, Sigma-Aldrich) and quantified using a Qubit Protein Assay Kit (Thermo Fisher Scientific). Samples were purified by TCA precipitation and processed with a filter-assisted sample preparation (FASP) protocol. First, 20 μg of protein was resuspended in 30 μl SDS denaturation buffer (4% SDS (w/v), 100 mM Tris–HCl, pH 8.2, 0.1 M DTT) and incubated at 95°C for 5 min. Then, samples were diluted with 200 μl UA buffer (8 M urea, 100 mM Tris–HCl, pH 8.2) and spun at 35°C at $14,000 \times g$ for 20 min in regenerated cellulose centrifugal filter units (Microcon 30, Merck Millipore). Samples were washed once with 200 μl of UA buffer and spun again at 35 °C and $14,000 \times g$ for 20 min. Cysteines were blocked with 100 μl IAA solution (0.05 M iodoacetamide in UA buffer) and incubated for 1 min at RT in a thermomixer at 600 rpm followed by $14,000 \times g$ centrifugation at 35°C for 15 min. Filter units were washed 3× with 100 μl of UA buffer and then 2× with a 0.5 M NaCl solution in water. Each wash step was followed by $14,000 \times g$ centrifugation at 35°C for 15 min. Proteins were digested overnight at room temperature with 1:50 ratio of trypsin (0.4 μg) in 130 μl TEAB (0.05 M triethylammonium bicarbonate in water). After digestion, peptide solutions were spun down at 35°C and $14,000 \times g$ for 15 min and acidified with 3 μl of 20% TFA (trifluoroacetic acid). Peptides were cleaned using Sep-Pak C18 silica columns (Waters Corporation) activated with 1 ml methanol and washed with a solution 1 ml of 60% ACN (acetonitrile) and 0.1% TFA. Columns were equilibrated with $3 \times 1$ ml of 3% ACN 0,1% TFA. The samples were diluted in 800 μl of 3% ACN 0.1% TFA and loaded onto the silica columns, then washed with $4 \times 1$ ml 3% ACN 0.1% TFA and eluted with 60% ACN 0.1% TFA. Samples were lyophilized in a SpeedVac and re-solubilized in 19 μl 3% ACN 0.1% FA (formic acid). 1 μl of synthetic peptides (Biognosys AG) was added to each sample for retention time calibration. Peptides were analysed by LC-MS/MS (Orbitrap Fusion™ Tribid™ MS, Functional Genomics Center Zürich). Samples were randomized, and group identities were unknown during measurements to blind the experimenter. Raw data were quantitatively and qualitatively analysed by Mascot and Progenesis QI platforms.

### C-reactive protein (CRP) ELISA

CRP concentration in serum was measured using a quantitative sandwich ELISA (Abcam, ab157712) according to the manufacturer's protocol. Briefly, 100 μl of standards, control and serum samples (diluted 1:10 (v/v) in supplied diluent) from adult MSUS and control males were added to wells containing anti-CRP. Then, HRP-conjugated anti-CRP antibodies were added. After 10 min of incubation at room temperature, wells were washed and then exposed to the chromogenic substrate tetramethylbenzidine (TMB) for 5 min in the dark at room temperature; then, a stop solution was added to each well. Absorbance was measured at 450 nm with a NOVOstar Plate Reader (BMG Labtech), and sample absorbance (directly related to concentration) was determined by plotting a standard curve with manufacturer-supplied standards and controls. Concentrations were analysed with one-tailed Student's *t*-test for mouse samples (as validation of proteomic data).

### Statistical analysis

Samples size was estimated based on our previous work with the MSUS model (Franklin *et al*, 2010, 2011; Gapp *et al*, 2014a,b, 2016a,b; Bohacek *et al*, 2015). Bodyweight, GC-1 spg luciferase

luminescence and PPARγ transcription factor binding assay were assessed using two-tailed Student's *t*-tests. CRP and aldosterone ELISA measurements were assessed using one-tailed Student's *t*-test since they were validation of proteomic and metabolomic datasets. For some phenotyping experiments, the data presented were reproduced from previously published findings, in which case one-tailed tests were used. In all such cases, this is clearly stated in the text. Glucose tolerance tests and glucose in response to restraint challenge were analysed using repeat-measures ANOVA and corrected for multiple comparisons using Šidák post hoc test. Most data matched the requirements for parametric statistical tests (normal distribution and homogeneity of variance). For data not normally distributed, Mann–Whitney *U*-test was used. This was the case for serum-injected offspring weight measurements, PLMS depression scale and PLMS age. Outliers were determined using the pre-defined criteria of adding and subtracting twice the standard deviation from the mean. Reported *n* represents number of animals after outlier removal. Whenever possible, re-analyses of data with *n* as litters instead of individual mice were conducted and confirmed the effects. Use of *n* as litters was considered but not implemented due to limited mouse numbers authorized by the animal license. Animals were randomly distributed to control and treatment groups prior to the experiments and before any contact with the experimenters. For individual metabolites and metabolomics enrichments, FDR was calculated using Benjamini–Hochberg (BH) post hoc test. Statistics were mainly computed with GraphPad Prism unless stated otherwise. Reported *n* for metabolomics represents biological replicates. Error bars represent s.e.m. in all figures. For all data, significance was set at a minimum $P < 0.05$. For a trend, $\#P < 0.1$. Asterisks represent significance as follows, $*P < 0.05$, $**P < 0.01$, $***P < 0.001$ and $****P < 0.0001$. Type of analysis and descriptive data for each statistical test are presented in figure legends.

## Data availability

The datasets collected in this study are available in the following databases:

- RNA-sequencing data: Gene Expression Omnibus GSE154369 (https://www.ncbi.nlm.nih.gov/geo/query/acc.cgi?acc=GSE154369)
- Raw data: ETH research collection 20.500.11850/426674 (https://doi.org/20.500.11850/426674)
- Proteomics data: ETH research collection 20.500.11850/426674 (https://doi.org/20.500.11850/426674)
- Metabolomics data: ETH research collection 20.500.11850/426674 (https://doi.org/20.500.11850/426674)

**Expanded View** for this article is available online.

## Acknowledgements

We thank Irina Lazar-Contes and Martin Roszkowski for assisting with MSUS breeding, Silvia Schelbert for taking care of the animal license and laboratory organization in Zürich, Catharine Aquino at the Functional Genomics Center Zürich (FGCZ) for assisting with RNA sequencing, Lukas von Ziegler, Paolo Nanni and Peter Gehrig for support with proteomics sample preparation and analysis, Johannes Bohacek for advice and Yvonne Zipfel for animal care in Zürich. We thank Paul Green for help with serum injections and Pawel Zielekinski for help with general animal care in Cambridge. We thank Darren Logan and Chris Lelliot for conceptual support and recommendations for tesaglitazar injections, and Wayo Matsushima and Tomas diDomenico for advice on early bioinformatics analysis. We are highly grateful to the administration of the SOS Children's Village, Pakistan, Saba Faisal, Rubina Asghar Ali, Almas Butt and Sajida Makhdoom at The Educators school, Lahore, Pakistan, for allowing the assessment of PLMS and control children, respectively, Anooshay Abid and Mehr Shafique at Lahore University of Management Sciences for technical help, Omar Chughtai at Chughtai Laboratories in Lahore for assistance with blood collection and Safeeullah Chaudhry and Shaper Mirza at Lahore University of Management Sciences for organizational support. We thank the University Zürich, the ETH Zürich, the Swiss National Science Foundation (31003A-135715), ETH grants (ETH-10 15-2 and ETH-17 13-2), Novartis Foundation (16B097), Roche Postdoctoral Fellowship Program (ID233), Cancer Research UK (C13474/A18583), the Escher Foundation and the Wellcome Trust (104640/Z/14/Z, 092096/Z10/Z). Katharina Gapp was supported by an early and advanced postdoc mobility fellowship from the Swiss National Science Foundation. Deepak K. Tanwar was supported by a Swiss Government Excellence Scholarship.

## Author contributions

GvS, KG and IMM conceived and designed the study. GvS and IMM wrote the manuscript with help from KG. GvS conducted MSUS treatments together with FM; collected and prepared plasma for metabolomic and proteomic analyses and tissue from MSUS and tesaglitazar-injected mice; performed molecular analyses of tissues; performed tesaglitazar, T0070907 and vehicle injections; organized breeding for tesaglitazar-, T0070907- and vehicle-injected mice; phenotyped offspring together with FM; and collected sperm from tesaglitazar-injected males used for sequencing. KG collected serum for injections, organized breeding, phenotyped serum-injected offspring and prepared RNA libraries from MSUS and control serum. GvS extracted and purified sperm RNA from tesaglitazar- and vehicle-injected males. KG prepared libraries from Day 46 sperm RNA, and GvS prepared libraries from Day 1 sperm RNA. AJ collected serum and saliva samples from children at the SOS village in Lahore, Pakistan, and performed all related data measurements including analysis of CES-DC results, performed the CRP ELISA and assisted with writing sections of the manuscript. P-LG performed bioinformatic analysis, helped to prepare figures, assisted with statistical analyses and provided key insight into manuscript development. FM organized animal housing and breeding logistics in Zürich, tracked animal welfare and performed phenotyping with GvS. DKT assisted P-LG with bioinformatics analyses. NZ measured metabolites in plasma, serum and saliva and analysed the data. NG, AE and KMT helped with molecular analyses. IMM provided essential conceptual support throughout the project, and IMM and EAM raised funds to finance the project.

## Conflict of interest

The authors declare that they have no conflict of interest.

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
