## [Review Process File · The EMBO Journal]

Involvement of circulating factors in the transmission of paternal experiences through the germline

Gretchen van Steenwyk, Katharina Gapp, Ali Jawaaid, Pierre-Luc Germain, Francesca Manuella, Deepak Tanwar, Nicola Zamboni, Niharika Gaur, Anastasiia Efimova, Kristina Thumfart, Eric Miska, and Isabelle Mansuy

DOI: [10.15252/embj.2020104579](https://doi.org/10.15252/embj.2020104579)

Corresponding authors: Isabelle Mansuy (mansuy@hifo.uzh.ch)

Review Timeline:

Submission Date:	29th Jan 20
Editorial Decision:	29th Jun 20
Revision Received:	27th Jul 20
Editorial Decision:	13th Aug 20
Revision Received:	4th Sep 20
Accepted:	16th Sep 20

Editor: Ieva Gailite

Transaction Report:

Thank you for submitting your manuscript for consideration by the EMBO Journal. We have now received two referee reports on your manuscript, which are included below for your information.

As you will see from the comments, while both reviewers find the study of interest, they also raise a number of issues that would have to be addressed and clarified before they can support publication of the manuscript. From my side, I find the reviewer comments generally reasonable. Therefore, I would like to invite you to submit a revised version of your manuscript in response to reviewers' comments.

I should add that it is The EMBO Journal policy to allow only a single major round of revision and that it is therefore important to resolve the main concerns at this stage. We are aware that many laboratories cannot function at full efficiency during the current COVID-19/SARS-CoV-2 pandemic, and I would be happy to discuss the revision in more detail via email or phone/videoconferencing.

We have extended our 'scooping protection policy' beyond the usual 3 month revision timeline to cover the period required for a full revision to address the essential experimental issues. This means that competing manuscripts published during revision period will not negatively impact on our assessment of the conceptual advance presented by your study. Please contact me if you see a paper with related content published elsewhere to discuss the appropriate course of action.

Please feel free to contact me if you have any further questions regarding the revision. Thank you for the opportunity to consider your work for publication. I look forward to receiving your revised manuscript.

Referee #1:

This manuscript describes experiments which address how the effects of trauma in males may modulate sperm epigenetics, identifying circulating factors as potential mediators. A logical series of experiments were performed. The experiments are generally well performed and appropriately analysed. Methods and statistical analyses as well as graphical presentation are valid. In summary, the findings are interesting, new, and significant. However, there are a number of ways in which the manuscript could be improved, as described below.

4. The health history did not appear to consider body weight/BMI, or alcohol intake. I would be expected that a substantial proportion of individuals in this age range would drink alcohol, and many might engage in binge drinking. Furthermore, was physical fitness by frequency of physical exercise was recorded?
5. Heterogeneous racial sampling is a potential concern, because of unequal representation of the ethnic groups. If the control samples have been compared with other published findings internationally, this is worth noting.
6. I am not sure if the guideline for this kind of article require this rather unusual style (i.e. no separation into introduction, results, discussion; lack of headings/subheadings), but I think it makes the manuscript hard to read. Additionally, the authors sometimes use long, complex sentences, which could be broken up or simplified, to help the reader.
7. The authors have performed a large number of analyses. It would be nice to get an overview at some point (maybe a schematic figure with all the applied methods and what exactly has been done with mice and what with human samples). Thus I suggest the authors add a schematic methodological overview/timeline and (if possible) subheadings.
8. From reading the abstract, I initially got the impression the work was only conducted in mice, and that the reference to human data was just background information. However, later on in the manuscript it becomes obvious that human data were also collected. I think this should be clarified in the abstract.
9. How can the authors to be sure that the serum injection causing the offspring phenotype change is due to the specific metabolites they found? Have they tried injecting the metabolites alone?
10. Do the authors think the injection of serum from WT mice to MS mice will reverse the phenotype? Can they do this experiment?
11. The authors should also discuss more about the limitations and future directions of this study. For example, do the authors predict that the F2 offspring will still show the same phenotype? Do they have data to address this question?
12. What is the potential role of extracellular vesicles as potential intermediaries in the 'circulation to sperm' effects described in this manuscript?
13. In Extended Figure 3A, the dot plot for the PLMS group appears as horizontal lines. This should be corrected so that individual dots are visible.

Referee #2:

In this study the investigators test the role of stress on the PPAR pathway in both a mouse model of unpredictable maternal stress (MSUS) as well as in children exposed to stress. This group as well as others have found metabolic and behavior dysfunction in mice transmitted across generations in response to stress and is now presenting work on mechanisms for transmission. Here, they pursue metabolic outcomes, with a focus on metabolomics and show a particular upregulation of polyunsaturated fatty acids metabolites in response to MSUS in plasma of adult males as well as

their offspring. Additionally, they identify a group of children exposed to stress and show a similar metabolomic profile, suggesting conservation of pathways. The group then identifies PPAR as mediating the effects from the fathers and use pharmacological activation and inhibition in mice to reproduce metabolic dysfunction. The conclusion of these and other experiments (sperm transcriptome) suggest a role for circulating metabolites in transmitting stress outcomes across generations.

The strengths of this interesting study include the correlation between the metabolic outcomes of human and mouse stress. Additionally, no prior studies have tested effect of direct perturbation of PPAR pathway on germline development and phenotypic transmission. The investigators were also able to test the correlation of sperm RNA-seq of Tesaglitazar-treated mice with previous studies of MSUS-treated mice. Nevertheless, there are both major and minor criticisms that should be addressed.

Major Criticisms

- The investigators provide no justification provided for only studying males - are females unaffected by MSUS (specifically the outcomes studied here)? If not, why were they excluded from analyses?
- The link between MSUS and PPAR pathway dysregulation is purely correlative. A much more compelling study would have been treatment of MSUS with PPAR antagonists to test for phenotypic rescue!! Also, it could also be argued that the intense focus on PPAR manipulation is a distraction from the central message of the study (circulating factors influence paternal phenotypic transmission).
- No justification provided for concentrations of Tesaglitazar (PPAR agonist) and T0070907 (PPAR antagonist) used in this study. Also, the investigators failed to validate that drug treatment produced a comparable effect to MSUS on PPAR stimulation in white adipose / liver / sperm.
- There is weak evidence for F2 phenotypic transmission (frequently unaffected). Also, related, what is "glucose response to restraint"? While the GTTs are unaffected, it is unclear what response to restraint is meant to show.

Minor Criticisms

- It would be desirable for the authors to provide an alternative mechanism addressing why there are no differences by RNA-seq in spermatogonial stem cells following Tesaglitazar treatment, but differences are apparent in transcriptionally quiescent sperm. It seems like the authors are trying to invoke somatic cells in the gonad or exosomal transport in the Discussion, but this needs to be clearer.
- Why researchers are studying PLMS saliva (not likely to effect gonadal health and development)?
- The strongest effects were observed for arachidonic acid metabolism, and yet researchers choose not to investigate the arachidonic / eicosanoid signaling pathway and its role in inflammation. Why?
- It is unclear how many mouse litters used in the study. Is there one mouse/litter?
- There is no discussion of how transposable element RNA dysregulation in sperm would impact offspring, and the correlation between affected linc/mRNA in MSUS and Tesa-injected mice is weak.
- The one-tailed t-test used in Extended Data Fig 2 doesn't seem justified.

Response to reviewers

Referee #1

This manuscript describes experiments which address how the effects of trauma in males may modulate sperm epigenetics, identifying circulating factors as potential mediators. A logical series of experiments were performed. The experiments are generally well performed and appropriately analysed. Methods and statistical analyses as well as graphical presentation are valid. In summary, the findings are interesting, new, and significant. However, there are a number of ways in which the manuscript could be improved, as described below.

1. One issue is the apparent lack of normalisation relative to the number of sperm cells. This directly affects the quantity of RNA isolated, and could lead to over (or under) representation of certain RNAs, and thus be a confound. Furthermore, there was no information provided regarding sperm counts of the individual donor at each time point, no information regarding range of sperm counts of individuals across time, no sperm count of mature sperm isolated for the RNA isolations, no normalisation to account for potential differences in sperm count prior to RNAseq.

Response: The reviewer is right that sperm count could influence the quantity of RNA isolated and eventually bias representation of certain RNAs if it is different across samples. We confirmed that sperm count is not altered by MSUS by measuring the total number of sperm cells in each sample (Figure 1A below). We also confirmed that sperm count is not altered in males treated with tesaglitazar both 1 day and 46 days after the end of treatment (Figure 1B below). Regarding RNA normalization, we routinely quantify the amount of RNA extracted from sperm and normalize it to use equal amount for preparing sequencing libraries. Therefore, all our sperm RNA sequencing data are controlled for and have no confound due to different sperm count or RNA amount. These data are now included in the manuscript as Extended Data Fig. 22.

Figure 1. No difference in sperm count in MSUS males or in males treated with tesaglitazar.

Number of sperm cells in A) Control (n=8) and MSUS (n= 8) males, 2-tailed Student's t-test, P=0.094, t=1.797, df=14, and in B) males treated with tesaglitazar, 1 day or 46 days after the final injection. Vehicle control-injected, n=6, Tesaglitazar-injected, n=6. 1 day after: 2-tailed Mann-Whitney test, P=0.31, Mann-Whitney U=11. 46 days after: 2-tailed Student's t-test, P=0.59, t=0.555, df=10.

3. The human cohort does not fully model the normative age group for males with reproductive intentions, so the relevance of this new information needs to be properly considered, with respect to age and lifestyle factors in this cohort. This could be briefly discussed.

Response: The reviewer is right that our SOS children cohort does not model a group at active reproduction age but instead, it models a group at an age of exposure. The purpose of having a cohort of children at exposure is to demonstrate that our findings in mouse pups exposed to trauma are validated and translatable to humans. Having a cohort at reproductive age would also be interesting but it would address a different question. It would also be more prone to confounding factors linked to lifestyle, which are minimized in our children cohort. Thus, the cohort is well suited to the study because all children have been exposed to a comparable type of trauma (paternal loss and maternal separation) at a comparable age, which helps eliminate confounds and facilitates correlative analyses with mouse data. Since all SOS children live in the same orphanage, lifestyle is quite similar, as well as the environment and eating habits. All these advantages make the cohort highly relevant. These points are now discussed in the revised manuscript.

4. The health history did not appear to consider body weight/BMI, or alcohol intake. I would be expected that a substantial proportion of individuals in this age range would drink alcohol, and many might engage in binge drinking. Furthermore, was physical fitness by frequency of physical exercise was recorded?

Response: The health history of our cohort indeed does consider body weight and BMI, we are sorry if the reviewer overlooked these data which are shown in Extended Data Fig 4b. Regarding alcohol intake, since the children are young (less than 13 years) and live in an orphanage under close supervision, they do not have access to alcohol. It is unlikely that they consume alcohol or engage in binge drinking. Alcohol consumption in Pakistan is actually banned for Muslims. Physical fitness and exercise were indirectly considered by grossly evaluating nutritional intake and access to playground facilities, which are comparable in SOS children since they live in the same orphanage and have a similar daily program.

5. Heterogeneous racial sampling is a potential concern, because of unequal representation of the ethnic groups. If the control samples have been compared with other published findings internationally, this is worth noting.

Response: Considering that our cohort is Pakistani and mostly includes Punjabi ethnicity, it is racially fairly homogeneous. This is advantageous and considerably reduces genetic diversity in the samples. Regarding previously published findings, it was shown that BMI, diet and ethnicity account for less than 5% of the variance between serum metabolites in healthy children from 6 different European populations (Lau et al., 2018), suggesting that our control samples are indeed comparable to other populations. This is now mentioned in the results section "Blood metabolites are altered in children exposed to early life trauma".

6. I am not sure if the guideline for this kind of article require this rather unusual style (i.e. no separation into introduction, results, discussion; lack of headings/subheadings), but I think it makes the manuscript hard to read. Additionally, the authors sometimes use long, complex sentences, which could be broken up or simplified, to help the reader.

Response: To make the manuscript easier to read, we introduced subsections and subheadings, and simplified/shortened sentences.

7. The authors have performed a large number of analyses. It would be nice to get an overview at some point (maybe a schematic figure with all the applied methods and what exactly has been

done with mice and what with human samples). Thus I suggest the authors add a schematic methodological overview/timeline and (if possible) subheadings.

Response: We thank the reviewer for suggesting to add a schematic overview of the experiments. This overview is now in Extended Data Fig. 23.

8. From reading the abstract, I initially got the impression the work was only conducted in mice, and that the reference to human data was just background information. However, later on in the manuscript it becomes obvious that human data were also collected. I think this should be clarified in the abstract.

Response: We thank the reviewer for suggesting to refer to the human data in the abstract. We modified the abstract accordingly.

9. How can the authors be sure that the serum injection causing the offspring phenotype change is due to the specific metabolites they found? Have they tried injecting the metabolites alone?

Response: We cannot be sure that phenotypic changes in the offspring are due to specific metabolites altered in MSUS serum. It is indeed unlikely that specific metabolites are responsible for the symptoms, but instead it is probably a combination of alterations in different metabolites together, possibly with alterations in other serum factors that are responsible. We have not tried injecting metabolites alone, because we would not know which ones to select since many are altered. Identifying which specific metabolites are altered would require precisely determining their absolute quantity in serum to derive the proper proportions to be injected for mimicking their alterations. But even if such information would be available, unfortunately, not all metabolites are commercially available, and we are not aware of any method for extracting specific metabolites from blood that would preserve their biological function in a way to be injectable. Further, while some metabolites are increased, others are decreased so injecting only some of those that are increased would only partially mimic the alterations, thus have low chance to produce any interpretable results.

10. Do the authors think the injection of serum from WT mice to MS mice will reverse the phenotype? Can they do this experiment?

Response: We do not think that the injection of control serum would reverse MSUS phenotypes. It is because it is improbable that MSUS symptoms can be reversed in adult animals since they are profoundly embedded in the body starting right after birth. Symptoms are induced during a critical period of development in early postnatal life (MSUS is from postnatal day 1 to 14) and likely involve permanent cellular and molecular anomalies. We know that MSUS alters the transcriptome, proteome, metabolome, methylome of different cells and tissues in adulthood and induces several complex symptoms. For instance, the dysregulated glucose/insulin metabolism likely involves pathways linked to insulin/glucagon secretion, glucose transporters, fatty acids and leptin regulation, etc, and different tissues e.g. pancreas, brain, liver, adipose tissue, and gonads. So it is difficult to conceive that injecting control serum in adults would have any counteracting effects against such widespread alterations induced many months earlier. Therefore, we believe that the hypothesis for conducting such experiment is not solid enough, considering the heavy and tedious experimental requirements it would involve (many animals, delicate i.v. injections, long-term analyses, etc). We are therefore not planning to do this experiment.

11. The authors should also discuss more about the limitations and future directions of this study. For example, do the authors predict that the F2 offspring will still show the same phenotype? Do they have data to address this question?

Response: As suggested by the reviewer, we now discuss more the limitations and future directions of the study in the discussion. Regarding the MSUS offspring, we previously showed that the F2 offspring of MSUS males have the same phenotypes as their father (Franklin et al., 2010; Gapp et al., 2014). In the present manuscript, we also show that the offspring and grand-offspring of tesaglitazar-injected males have altered weight and dysregulated blood glucose (Extended Data Fig. 8) similarly to their father.

12. What is the potential role of extracellular vesicles as potential intermediaries in the 'circulation to sperm' effects described in this manuscript?

Response: The role of extracellular vesicles as potential intermediates in circulation to the effects in sperm is not known but this is an interesting question. Extracellular vesicles are known mostly for carrying RNA and proteins in circulation (Doyle & Wang, 2019), and can act as communication vectors between cells and tissues (Thomou et al., 2017). It is therefore possible that they reach gonads and transfer their content to testicular cells including germ cells. In the case of MSUS and tesaglitazar injection, extracellular vesicles may therefore participate to changes observed in sperm. Isolation and purification of extracellular vesicles from blood and their targeted analysis would be necessary to determine if they are altered by MSUS or not. This is now discussed in the manuscript.

13. In Extended Figure 4A, the dot plot for the PLMS group appears as horizontal lines. This should be corrected so that individual dots are visible.

Response: We thank the reviewer for pointing out this error, it has now been corrected.

References

- Doyle, L. M., & Wang, M. Z. (2019). Overview of extracellular vesicles, their origin, composition, purpose, and methods for exosome isolation and analysis. *Cells*, *8*(7). <https://doi.org/10.3390/cells8070727>
- Franklin, T. B., Russig, H., Weiss, I. C., Gräff, J., Linder, N., Michalon, A., ... Mansuy, I. M. (2010). Epigenetic transmission of the impact of early stress across generations. *Biological Psychiatry*, *68*(5), 408–415. <https://doi.org/10.1016/j.biopsych.2010.05.036>
- Gapp, K., Jawaid, A., Sarkies, P., Bohacek, J., Pelczar, P., Prados, J., ... Mansuy, I. M. (2014). Implication of sperm RNAs in transgenerational inheritance of the effects of early trauma in mice. *Nature Neuroscience*, *17*(May), 667–669. <https://doi.org/10.1038/nn.3695>
- Lau, C.-H. E., Siskos, A. P., Maitre, L., Robinson, O., Athersuch, T. J., Want, E. J., ... Coen, M. (2018). Determinants of the urinary and serum metabolome in children from six European populations. *BMC Medicine*, *16*(1), 202. <https://doi.org/10.1186/s12916-018-1190-8>
- Thomou, T., Mori, M. A., Dreyfuss, J. M., Konishi, M., Sakaguchi, M., Wolfrum, C., ... Kahn, C. R. (2017). Adipose-derived circulating miRNAs regulate gene expression in other tissues. *Nature*, *542*(7642), 450–455. <https://doi.org/10.1038/nature21365>

Referee #2

In this study the investigators test the role of stress on the PPAR pathway in both a mouse model of unpredictable maternal stress (MSUS) as well as in children exposed to stress. This group as well as others have found metabolic and behavior dysfunction in mice transmitted across generations in response to stress and is now presenting work on mechanisms for transmission. Here, they pursue metabolic outcomes, with a focus on metabolomics and show a particular upregulation of polyunsaturated fatty acids metabolites in response to MSUS in plasma of adult males as well as their offspring. Additionally, they identify a group of children exposed to stress and show a similar metabolomic profile, suggesting conservation of pathways. The group then identifies PPAR as mediating the effects from the fathers and use pharmacological activation and inhibition in mice to reproduce metabolic dysfunction. The conclusion of these and other experiments (sperm transcriptome) suggest a role for circulating metabolites in transmitting stress outcomes across generations.

The strengths of this interesting study include the correlation between the metabolic outcomes of human and mouse stress. Additionally, no prior studies have tested effect of direct perturbation of PPAR pathway on germline development and phenotypic transmission. The investigators were also able to test the correlation of sperm RNA-seq of Tesaglitazar-treated mice with previous studies of MSUS-treated mice. Nevertheless, there are both major and minor criticisms that should be addressed.

Response: We thank the reviewer for finding our study interesting and are happy to address her/his criticisms.

Major Criticisms

- The investigators provide no justification provided for only studying males - are females unaffected by MSUS (specifically the outcomes studied here)? If not, why were they excluded from analyses?

Response: The reviewer is asking if females were studied and if there are affected. Yes, we studied MSUS females in the past and found that they have normal weight but altered behaviors. For instance, MSUS females have depressive-like symptoms and increased risk-taking behaviors (Franklin et al., 2010; Weiss, Franklin, Vizi, & Mansuy, 2011). For the current study, we measured weight in MSUS female offspring and found that it is not altered (Extended Data Fig. 3).

- The link between MSUS and PPAR pathway dysregulation is purely correlative. A much more compelling study would have been treatment of MSUS with PPAR antagonists to test for phenotypic rescue!! Also, it could also be argued that the intense focus on PPAR manipulation is a distraction from the central message of the study (circulating factors influence paternal phenotypic transmission).

Response: The reviewer considers that the link between MSUS and PPAR pathways is purely correlative but this is partly true since we provide evidence that treatment of spermatogonia-like cells with MSUS blood leads to PPAR activation, suggesting a causal link between MSUS and PPAR in germ cells. The suggestion to treat MSUS mice with a PPAR antagonist to test for a phenotypic rescue is an interesting idea but this experiment is conceptually problematic and its outcome unlikely to show a reversal. This experiment is based on the premise that MSUS symptoms can be reversed in adult animals but this is improbable because the symptoms are induced during a critical period of development in early postnatal life (MSUS is from postnatal day 1 to 14) and likely involve permanent cellular and molecular anomalies, likely profoundly embedded into the body. We indeed know that MSUS alters the transcriptome, proteome, metabolome, methylome of different cells and tissues in adulthood and induces many symptoms, from metabolic to behavioral in adult animals. PPAR pathways are only one among many other

pathways involved in MSUS symptoms and in blood, multiple metabolic pathways are affected in adulthood. Therefore, it is difficult to conceive that injecting a PPAR antagonist alone in adults would have any counteracting effects against such widespread alterations.

Further, we show in the manuscript that injecting the PPAR antagonist T0070907 in adult control males at a dosage that is tolerable does not induce any effect on weight and glucose tolerance in the offspring (Extended Data Fig 12). We indeed tried injecting the drug to MSUS fathers in a pilot experiment a few years ago but this experiment failed to provide any conclusive results. Only 2 viable offspring pups could be obtained from 5 MSUS males injected with the antagonist, while 10-22 pups were obtained from (5-8) saline- or PPAR antagonist-injected controls or saline-injected MSUS fathers. This low pups yield may be due to combined detrimental effects of drug and MSUS on fertility (PPAR has been reported to be involved in fertility). Besides, PPAR antagonist had no effect on MSUS offspring weight (Fig. 1 below), suggesting that it cannot reverse MSUS symptoms. We hope that these negative pilot results will convince the reviewer that injecting a PPAR antagonist in MSUS mice will not provide any interpretable outcome and therefore that such long-term experiment is disproportionately difficult and not justified.

We would like to emphasize that our manuscript already includes several important findings and a massive amount of data that goes far beyond any standard in the field. It reports 3 different *in vivo* manipulations including MSUS, serum injection and tesaglitazar treatment, it includes data in mouse offspring and grand-offspring, corresponding data in humans, and provides transcriptomic, metabolomic and proteomic data in blood and transcriptomic data in sperm. We hope that the reviewer will appreciate that all these data and the collected results are largely sufficient for the manuscript and are strong enough to support the conclusions of the paper.

Figure 1. PPAR antagonist injection in MSUS and control fathers produces no effect on offspring weight. Offspring from saline-injected controls n=10, Offspring from antagonist-injected controls n=22, Offspring from saline-injected MSUS males n=15, Offspring from antagonist-injected MSUS males n=2. Two-way ANOVA for effect of MSUS $P=0.0088$, $F(1,45)=7.507$, for effect of drug $P=0.7661$, $F(1,45)=0.08956$. For comparison between individual groups, p-values reported on the graph are adjusted using Sidak's multiple testing correction.

- No justification provided for concentrations of Tesaglitazar (PPAR agonist) and T0070907 (PPAR antagonist) used in this study. Also, the investigators failed to validate that drug treatment produced a comparable effect to MSUS on PPAR stimulation in white adipose / liver / sperm.

Response: We apologize for not explaining our choice of concentration for tesaglitazar and T0070907. This choice was based on previous reports in the literature using these drugs. Several studies in mice used tesaglitazar at 20 ug/kg administered orally (Zadelaar et al., 2006; Zhang et al., 2012), and a comparable dose was used in humans (11-15 ug/kg) for intravenous delivery

(Ericsson et al., 2004). We therefore delivered 8 repeated injections each at a dose of 10 ug/kg in mice. For T0070907, a dosage of 2 mg/kg was used in rats for i.p. injection, therefore we delivered 2 i.p. injections (one per week for 2 weeks) at a dosage of 1.5 mg/kg in our mice.

Regarding validation that drug treatment produces comparable effects to MSUS, we did conduct comparative analyses in adipose tissue, liver and sperm. Tesaglitazar has been shown to increase PPAR α expression in white adipose tissue (Glinghammar et al., 2011), similar to the increase in PPAR γ activity in MSUS epididymal white adipose tissue (Extended Data Fig. 7a). In liver, we confirmed that genes known to be regulated by PPAR agonists in general, including *Abca1*, *Cpt1a* and *TNF a*, are differentially regulated in MSUS mice (Extended Data Fig. 7b). These results confirm altered PPAR signaling in MSUS tissue, leading to our downstream investigation of PPAR agonist injections. For the purpose of this paper, we believe that comparing somatic tissues in exposed animals is not the most relevant to transmission mechanisms. Therefore, we focused our comparison on sperm, which is directly link to the offspring. Comparative analyses of RNA-seq data from sperm of MSUS and tesaglitazar-treated males showed a significant correlation. There is also a significant correlation for the effects of MSUS and tesaglitazar on offspring metabolic parameters, including body weight and blood glucose in response to a glucose tolerance test (Fig. 3b-c).

- There is weak evidence for F2 phenotypic transmission (frequently unaffected). Also, related, what is "glucose response to restraint"? While the GTTs are unaffected, it is unclear what response to restraint is meant to show.

Response: We are sorry if the reviewer overlooked our F2 (grand-offspring) data but we do show that grand-offspring of tesaglitazar treated males have phenotypes comparable to their grand-father and father, in particular reduced weight and decreased glucose during a glucose tolerance test (Fig. 3, Extended Data Fig. 8). Data on the grand-offspring of MSUS fathers have been previously published (Franklin, Linder, Russig, Thöny, & Mansuy, 2011; Franklin et al., 2010; Gapp, Jawaid, et al., 2014; Gapp, Soldado-Magraner, et al., 2014). Our MSUS and tesaglitazar models are indeed among the rare models in the literature to have transgenerational effects observed until at least the 3rd generation. Please note that we refer to what the reviewer calls F2 as F3 in our previous publications.

Regarding glucose response to restraint, this is the increase in the level of glucose in blood induced when an animal/organism is exposed to stress, in this case restraint in a tight tube. It measures the capacity of the organism to respond to stress by mobilizing energy via glucose release. This response is different to GTT because GTT measures blood glucose response after injection of glucose, which evaluates whether glucose homeostasis/metabolism is properly maintained in the body.

Minor Criticisms

- It would be desirable for the authors to provide an alternative mechanism addressing why there are no differences by RNA-seq in spermatogonial stem cells following Tesaglitazar treatment, but differences are apparent in transcriptionally quiescent sperm. It seems like the authors are trying to invoke somatic cells in the gonad or exosomal transport in the Discussion, but this needs to be clearer.

Response: The reviewer is asking for an alternative mechanism for RNA-seq data in spermatogonial stem cells but the manuscript does not contain any analyses in spermatogonial stem cells, hence we cannot compare them to other datasets. As recommended by the reviewer, we now discuss the potential role of somatic cells and exosomal transport in the effects on sperm RNA.

- Why researchers are studying PLMS saliva (not likely to effect gonadal health and development)?

Response: We study PLMS saliva because saliva is an easily accessible body fluid in humans that can be collected non-invasively. Saliva is not expected to affect gonadal health and development but it is classically used as an additional means to measure physiological/metabolic parameters.

- The strongest effects were observed for arachidonic acid metabolism, and yet researchers choose not to investigate the arachidonic / eicosanoid signaling pathway and its role in inflammation. Why?

Response: The reviewer is right that arachidonic acid pathways (and eicosanoid signaling in humans) are among the most affected in our metabolomic profiling. However, we selected PPAR signaling for further analyses because i) it is highly relevant to metabolism which is altered in our MSUS model across generations, ii) research tools are available e.g. agonists and antagonists to manipulate PPAR signaling, iii) it is clinically relevant considering the use of PPAR drugs as triglycerides lowering agents in obese and diabetic patients (Grygiel-Górniak et al., 2014). Nonetheless, we did examine some aspects relevant to inflammation. For instance, we show that C-reactive protein is reduced in blood (proteomic data confirmed by ELISA, Extended Data Fig. 18b) suggesting that inflammatory pathways are affected, as suggested by the reviewer.

- It is unclear how many mouse litters used in the study. Is there one mouse/litter?

Response: We typically use several mice per litter and several different litters (up to 20 per group). The total number of male breeders used to generate the offspring and the number of offspring used for analyses are now provided for each experiment in Extended Data Fig. 20.

- There is no discussion of how transposable element RNA dysregulation in sperm would impact offspring, and the correlation between affected linc/mRNA in MSUS and Tesa-injected mice is weak.

Response: We thank the reviewer for these points and have now addressed the question of transposable element dysregulation in sperm on the offspring and the possible correlation between linc/mRNA and tesa-injected in the discussion.

- The one-tailed t-test used in Extended Data Fig 2 doesn't seem justified.

Response: We believe the use of one-tailed t-test is justified for Extended Data Fig. 2, since this assay is a validation by ELISA of a decrease in the level of aldosterone that was detected by mass spectrometry (TOF-MS). Since the expected change is a decrease (thus in one clearly defined direction), one-tailed t-test can be used. We hope that the reviewer will agree with us.

References

- Ericsson, H., Hamrén, B., Bergstrand, S., Elebring, M., Fryklund, L., Heijer, M., & Öhman, K. P. (2004). Pharmacokinetics and metabolism of tesaglitazar, a novel dual-acting peroxisome proliferator-activated receptor α/γ agonist, after a single oral and intravenous dose in humans. *Drug Metabolism and Disposition*, 32(9), 923-929.
- Franklin, T. B., Linder, N., Russig, H., Thöny, B., & Mansuy, I. M. (2011). Influence of early stress on social abilities and serotonergic functions across generations in mice. *PLoS One*, 6(7), e21842. <https://doi.org/10.1371/journal.pone.0021842>
- Franklin, T. B., Russig, H., Weiss, I. C., Gräff, J., Linder, N., Michalon, A., ... Mansuy, I. M. (2010). Epigenetic transmission of the impact of early stress across generations. *Biological Psychiatry*, 68(5), 408–415. <https://doi.org/10.1016/j.biopsych.2010.05.036>
- Gapp, K., Jawaid, A., Sarkies, P., Bohacek, J., Pelczar, P., Prados, J., ... Mansuy, I. M. (2014). Implication of sperm RNAs in transgenerational inheritance of the effects of early trauma in mice. *Nature Neuroscience*, 17(May), 667–669. <https://doi.org/10.1038/nn.3695>
- Gapp, K., Soldado-Magraner, S., Alvarez-Sanchez, M., Bohacek, J., Vernaz, G., Shu, H., ... Mansuy, I. M. (2014). Early life stress in fathers improves behavioural flexibility in their offspring. *Nature Communications*, 5, 5466. <https://doi.org/10.1038/ncomms6466>

- Glinghammar, B., Berg, A.-L., Bjurström, S., Stockling, K., Blomgren, B., Westerberg, R., ... Andersson, U. (2011). Proliferative and molecular effects of the dual PPARalpha/gamma agonist tesaglitazar in rat adipose tissues: relevance for induction of fibrosarcoma. *Toxicologic Pathology*, 39(2), 325–336. <https://doi.org/10.1177/0192623310394210>
- Grygiel-Górniak, B., Berger, J., Moller, D., Boitier, E., Gautier, J., Roberts, R., ... Herz, M. (2014). Peroxisome proliferator-activated receptors and their ligands: nutritional and clinical implications – a review. *Nutrition Journal*, 13(1), 17. <https://doi.org/10.1186/1475-2891-13-17>
- Weiss, I. C., Franklin, T. B., Vizi, S., & Mansuy, I. M. (2011). Inheritable effect of unpredictable maternal separation on behavioral responses in mice. *Frontiers in Behavioral Neuroscience*, 5(February), 3. <https://doi.org/10.3389/fnbeh.2011.00003>
- Zadelaar, A. S. M., Boesten, L. S. M., Jukema, J. W., van Vlijmen, B. J. M., Kooistra, T., Emeis, J. J., ... Havekes, L. M. (2006). Dual PPARalpha/gamma agonist tesaglitazar reduces atherosclerosis in insulin-resistant and hypercholesterolemic ApoE*3Leiden mice. *Arteriosclerosis, Thrombosis, and Vascular Biology*, 26(11), 2560–2566. <https://doi.org/10.1161/01.ATV.0000242904.34700.66>
- Zhang, B.-C., Li, W.-M., Li, X.-K., Zhu, M.-Y., Che, W.-L., & Xu, Y.-W. (2012). Tesaglitazar ameliorates non-alcoholic fatty liver disease and atherosclerosis development in diabetic low-density lipoprotein receptor-deficient mice. *Experimental and Therapeutic Medicine*, 4(6), 987–992. <https://doi.org/10.3892/etm.2012.713>

Thank you for submitting a revised version of your manuscript. Your study has now been seen by both original referees, who find that most of their main concerns have been addressed and are now broadly in favour of publication of the manuscript. There now remain only a few mainly editorial issues that have to be addressed before I can extend formal acceptance of the manuscript.

Referee #1:

The manuscript has been extensively revised and my comments adequately addressed.

Referee #2:

The authors addressed many of the questions raised by the two reviewers. The argument in favor of PPAR signaling for conferring the MSUS phenotype from fathers to sons is compelling. It is also consistent given that pharmacological activation of PPAR in vivo can reproduce metabolic dysfunction in the offspring. Nevertheless, even though it is not a trivial experiment to perform (importance of PPAR for development and lack of a suitable antagonist), it still would have been desirable to see a rescue of the MSUS phenotype if activation of PPAR was sufficient to confer the phenotype.

The authors performed the requested editorial changes.

Editor accepted the manuscript.

Corresponding Author Name: Isabelle M Mansuy

Manuscript Number: EMBOJ2020104579R